# The antimicrobial potential of cannabidiol

Mark A. T. Blaskovich [1✉], Angela M. Kavanagh[1], Alysha G. Elliott [1], Bing Zhang [1], Soumya Ramu[1],
Maite Amado[1], Gabrielle J. Lowe[1], Alexandra O. Hinton[1], Do Minh Thu Pham[1], Johannes Zuegg[1], Neil Beare[2],
Diana Quach[3], Marc D. Sharp[3], Joe Pogliano[3,4], Ashleigh P. Rogers[5], Dena Lyras [5], Lendl Tan [6],
Nicholas P. West [6], David W. Crawford[7], Marnie L. Peterson[7], Matthew Callahan[8] & Michael Thurn[8]

Antimicrobial resistance threatens the viability of modern medicine, which is largely dependent on the successful prevention and treatment of bacterial infections. Unfortunately, there are few new therapeutics in the clinical pipeline, particularly for Gram-negative bacteria. We now present a detailed evaluation of the antimicrobial activity of cannabidiol, the main non-psychoactive component of cannabis. We confirm previous reports of Gram-positive activity and expand the breadth of pathogens tested, including highly resistant *Staphylococcus aureus*, *Streptococcus pneumoniae*, and *Clostridioides difficile*. Our results demonstrate that cannabidiol has excellent activity against biofilms, little propensity to induce resistance, and topical in vivo efficacy. Multiple mode-of-action studies point to membrane disruption as cannabidiol's primary mechanism. More importantly, we now report for the first time that cannabidiol can selectively kill a subset of Gram-negative bacteria that includes the 'urgent threat' pathogen *Neisseria gonorrhoeae*. Structure-activity relationship studies demonstrate the potential to advance cannabidiol analogs as a much-needed new class of antibiotics.

[1] Centre for Superbug Solutions, Institute for Molecular Bioscience, The University of Queensland, St. Lucia, QLD 4072, Australia. [2] BDG Synthesis, Wellington 5045, New Zealand. [3] Linnaeus Bioscience Inc., 3210 Merryfield Row, San Diego, CA 92121, USA. [4] Division of Biological Sciences, University of California, San Diego, CA 92093, USA. [5] Infection and Immunity Program, Monash Biomedicine Discovery Institute and Department of Microbiology, Monash University, Clayton, VIC 3800, Australia. [6] School of Chemistry and Molecular Biosciences, The University of Queensland, St Lucia, QLD 4072, Australia. [7] Perfectus Biomed, LLC (formerly Extherid Biosciences), 3545 S Park Dr, Jackson, WY 83001, USA. [8] Botanix Pharmaceuticals Ltd., Level 1, 50 Angove Street, North Perth, WA 6005, Australia. ✉email: m.blaskovich@uq.edu.au

**B**acteria have become increasingly resistant to antibiotics, leading to the United States (US) Centers for Disease Control and Prevention (CDC) stating in a 2019 report that we need to "stop referring to a coming post-antibiotic era—it's already here"[1]. This "post-antibiotic era" means that millions of people who are at increased risk of infection due to surgery, cancer care, organ transplant, kidney dialysis or chronic conditions such as diabetes face a potentially untreatable illness. Unfortunately, the unfavorable economics of antibiotic development[2–4] means that almost all major pharmaceutical companies have abandoned antibiotic research and placed smaller biotech companies in a precarious financial position, leaving few new therapeutics in the clinical pipeline[5–7]. This is particularly true for Gram-negative infections, where there have essentially been no novel molecular classes discovered and approved for clinical use since the 1960's, but we also need better Gram-positive therapies, as resistant Gram-positive infections still cause substantial mortality[1].

Cannabidiol (CBD), the main non-psychoactive ingredient of the cannabis plant, is a small molecule (MW 314 Da) phytocannabinoid consisting of a pentyl-substituted bis-phenol aromatic group (pentylresorcinol) linked to an alkyl-substituted cyclohexene terpene ring system (Fig. 1a). It is one of more than 100 cannabinoids that can be extracted from the *Cannabis sativa*

L. plant, many of which have been shown to be biologically active[8]. CBD was first isolated from Minnesota Wild Hemp in 1940[9], but its structure was not completely elucidated until 1963[10]. CBD has quite a remarkable polypharmacology, and has been extensively tested for a variety of disease indications. Cannabinoid clinical studies have examined reduction of chemotherapy-induced nausea and vomiting, appetite stimulation in HIV/AIDS, and treatment of chronic pain, spasticity due to multiple sclerosis or paraplegia, depression, anxiety disorder, sleep disorder, psychosis, glaucoma, and Tourette syndrome[11]. CBD has anti-inflammatory[12] and neuroprotective[13] properties. A highly purified oil-based liquid formulation of CBD (Epidiolex® in the US and Epidyolex in the European Union [EU]) was approved in 2018 by the Food and Drug Administration (FDA) and in 2019 by the European Medicines Agency (EMA) for the oral treatment of two epilepsy disorders, Dravet syndrome and Lennox-Gastaut syndrome[14–16]. However, unregulated CBD oil products are also widely used by the public, with a variety of concerns over legality, quality, and safety[17], particularly in children[18], where CBD was used to treat Dravet syndrome before official clinical approval[19].

Among CBD's many reported pharmacological properties is antimicrobial activity, with a 1976 publication by van Klingeren and ten Ham[20] reporting minimum inhibitory concentrations (MICs)

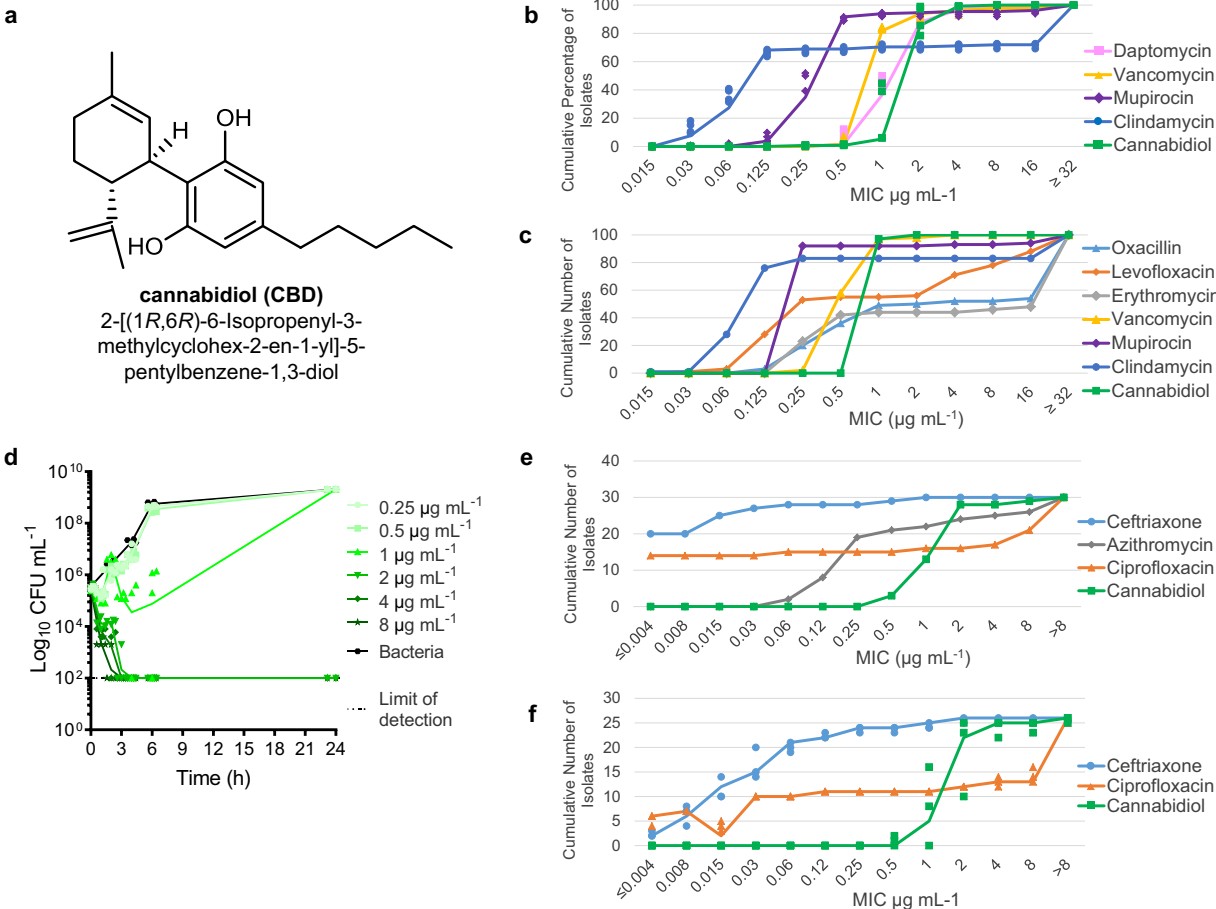

**Fig. 1 Antibacterial activity of cannabidiol. a** Structure of CBD. **b** MIC$_{90}$ distribution of CBD against 132 Australian *S. aureus* isolates. Cumulative percentage of isolates below or equal to a given MIC are indicated. Data are $n = 4$ biologically independent samples for MIC of each isolate. **c** MIC$_{90}$ distribution of CBD against 100 USA *S. aureus* isolates. Cumulative number of isolates below or equal to a given MIC are indicated. Data are $n = 1$ for MIC of each isolate. **d** Time-kill assay of CBD against MRSA ATCC 43300. Data are mean ± SD for $n = 4$ biologically independent samples. **e** Broth microdilution MIC distribution of CBD against 30 *N. gonorrhoeae* isolates. Cumulative number of isolates below or equal to a given MIC are indicated. Data are $n = 1$. **f** Agar microdilution MIC distribution of CBD against 26 *N. gonorrhoeae* isolates. Cumulative number of isolates below or equal to a given MIC are indicated. Data are $n = 3$ independent experiments.

for both purified extracted CBD and $\Delta^9$-tetrahydrocannabinol (THC) in the range of 1–5 µg mL$^{-1}$ for Gram-positive staphylococci and streptococci. Both cannabinoids were largely inactivated in media containing 4% serum (MIC 50 µg mL$^{-1}$) and did not show activity against Gram-negative *Escherichia coli*, *Salmonella* Typhi, or *Proteus vulgaris* (MIC >100 µg mL$^{-1}$). A number of earlier[21] and subsequent[22] publications have reported on the antimicrobial properties of cannabis extracts, as opposed to purified compounds. The initial description of CBD's antibacterial activity appears to have been largely ignored until 2008, when a new report[23] assessed the potency of all five major cannabinoids (CBD, cannabichromene [CBC], cannabigerol [CBG], THC and cannabinol [CBN]) against six methicillin-resistant *Staphylococcus aureus* (MRSA) strains of current clinical relevance: all showed MIC values in the 0.5–2 µg mL$^{-1}$ range. A number of synthetic derivatives were also tested. A recent report has provided additional antimicrobial characterization of CBG[24].

CBD is undergoing Phase 2 clinical trials for the topical treatment of acne (NCT03573518) and atopic dermatitis (NCT03824405), based on its reported anti-inflammatory properties. Given that both skin diseases are also associated with bacterial infections, we were interested to undertake a more detailed examination of CBD's antimicrobial properties. A key distinction of these current studies is that they are based on synthetic CBD, rather than the purified extracted CBD used in previous reports, avoiding the potential for batch-to-batch variability due to different plant sources, which may lead to distortion of results due to minor but potentially potent biologically-derived impurities. Our studies demonstrate that CBD has consistent activity against a wide range of Gram-positive bacteria, including a variety of drug-resistant strains. Intriguingly, this activity extends to a small subset of Gram-negative bacteria, including pathogens of concern such as *Neisseria gonorrhoeae*. Preliminary structure-activity relationship studies indicate that there is scope for improving the antibiotic properties of CBD, with further research needed to develop analogs with systemic in vivo activity.

## Results

**Antimicrobial activity of cannabidiol.** We now report that CBD has some remarkably useful antimicrobial activity beyond that previously described[20,21,23], with potential clinical utility for nasal decolonization. It shows a strikingly consistent MIC of 1–4 µg mL$^{-1}$ against a diverse range of over 20 types of Gram-positive bacteria, including multiple strains of the key pathogens MRSA, multidrug-resistant (MDR) *Streptococcus pneumoniae*, *Enterococcus faecalis*, and the anaerobic bacteria *Clostridioides* (previously *Clostridium*) *difficile* and *Cutibacterium* (previously *Propionibacterium*) *acnes* (see Table 1 for key results, Supplementary Table 1 for complete results). Notably, the MIC did not appreciably change against highly resistant strains such as vancomycin-resistant *S. aureus* (VRSA), vancomycin-resistant enterococci (VRE) and the hypervirulent ribotype 027 strain of *C. difficile*, remaining in the range of 1–4 µg mL$^{-1}$. The MIC$_{90}$ against 132 MRSA and methicillin-susceptible *S. aureus* (MSSA) ATCC strains and Australian clinical isolates was 4 µg mL$^{-1}$ (compared to 2 µg mL$^{-1}$ for vancomycin, 4 µg mL$^{-1}$ for daptomycin, and 64 µg mL$^{-1}$ for clindamycin) (Fig. 1b, Supplementary Table 2). Additional testing against 50 MSSA and 50 MRSA USA-derived isolates showed CBD with an MIC$_{50}$ and MIC$_{90}$ of 1 µg/mL for all *S. aureus* (97/100 isolates): this panel had 17% resistance to clindamycin, 56% resistance to erythromycin, and 44% resistance to levofloxacin (Fig. 1c, Supplementary Tables 3, 4). CBD was less potent against a set of 59 beta-hemolytic streptococci (29 *S. pyogenes* and 30 *S. agalactiae*) with MIC values from 8–32 µg/mL and MIC$_{50}$ and MIC$_{90}$ of 16 µg/mL (Supplementary Tables 3, 5).

A time-kill assay against MRSA ATCC 43300 showed rapid bactericidal activity (<3 h), with a minimum bactericidal concentration (MBC) of 2 µg mL$^{-1}$ (Fig. 1d). In line with previous reports, addition of 50% human serum to the assay media abrogated antibacterial activity in our tests (MRSA MIC >256 µg mL$^{-1}$). This decreased activity is likely due to low levels of free CBD as a result of CBD's previously reported high levels of protein binding. One publication describes CBD as 86–90% bound in human plasma[25], though THC and other cannabinoids have even higher levels of 95–99% binding, primarily to lipoproteins[26,27], and the Epidiolex FDA filing indicates >94% protein binding for CBD and its metabolites (with up to >99% depending on method used)[28]. However, when 5% lysed horse blood (LHB) was added to supplemented Brucella broth (SBB) during our *C. acnes* MIC testing, MICs were not as dramatically impacted, rising from 1–2 µg mL$^{-1}$ to 8–16 µg mL$^{-1}$.

We have previously reported that microtiter plate composition can affect broth microdilution (BMD) MIC values[29], and indeed the MIC of CBD against MRSA ATCC 43300 in polystyrene plates (MIC 2 µg mL$^{-1}$) was substantially increased in polypropylene plates (MIC 16 µg mL$^{-1}$) and dramatically increased in non-binding surface plates (MIC >64 µg mL$^{-1}$). The agar MIC against MRSA ATCC 43300, as measured by serial dilution in Tryptic Soy Broth (TSB) with 1% agarose, was 1 µg mL$^{-1}$. Agar disc diffusion assays required 35 µg of CBD added to a 6 mm paper disc before any zone of inhibition was observed, with an 8 mm zone at 50 µg, a 9 mm zone at 100 µg, and a 10 mm zone at 150 µg (compared to a 14 mm zone for 15 µg of vancomycin, reaching 21 mm at 150 µg). CBD showed modest activity against *Mycobacterium smegmatis* (MIC 16 µg mL$^{-1}$) but marginal activity against *M. tuberculosis* H37Rv (MIC >64 µg mL$^{-1}$, though with 70% inhibition at 64 µg mL$^{-1}$) in BMD MIC assays. CBD was not active against the yeast strains tested, with MIC > 64 µg mL$^{-1}$ against *Candida albicans* and *Cryptococcus neoformans* (Supplementary Table 6).

CBD was also generally inactive against 20 species of Gram-negative bacteria (such as the key ESKAPE pathogens *E. coli*, *Klebsiella pneumoniae*, *Pseudomonas aeruginosa* and *Acinetobacter baumannii*), consistent with previous reports (see Table 1 for key results, Supplementary Table 7 for complete results). Surprisingly, it possessed excellent potency against four Gram-negative bacteria, including the dangerous pathogens *Neisseria gonorrhoeae* (MIC 1–2 µg mL$^{-1}$), *Neisseria meningitides* (MIC 0.25 µg mL$^{-1}$), *Moraxella catarrhalis* (MIC 1 µg mL$^{-1}$) and *Legionella pneumophila* (MIC 1 µg mL$^{-1}$). These bacteria are responsible for sexually transmitted gonorrhea, life-threatening meningitis, airway infections such as bronchitis and pneumonia, and Legionnaires' disease, respectively. Anti-gonorrheal activity was confirmed by an MIC$_{90}$ against 30 *N. gonorrhoeae* isolates comprised of reference strains from the WHO, isolates from the CDC antibiotic resistance bank, and clinical isolates, 16 of which were ciprofloxacin-resistant, ten azithromycin-resistant, and two ceftriaxone-resistant (Supplementary Table 8). The MIC$_{50}$/MIC$_{90}$ (determined in broth microdilution assays) for CBD was 2/2 µg mL$^{-1}$, whereas the comparators azithromycin, ciprofloxacin and ceftriaxone were 0.25/>8, 0.06/>8, and ≤0.008/0.03 µg mL$^{-1}$, respectively. CBD showed a much narrower distribution of MICs (Fig. 1e). Similar MIC values were obtained for a subset of 26 isolates when assayed under agar dilution assay conditions (Supplementary Table 8): MIC$_{50}$/MIC$_{90}$ for CBD was 2/4 µg mL$^{-1}$, ciprofloxacin 4/>8 µg mL$^{-1}$ and ceftriaxone 0.03/0.25 µg mL$^{-1}$ (Fig. 1f).

CBD was not active against efflux pump mutants of *E. coli* or *P. aeruginosa* (MIC >128 µg mL$^{-1}$), meaning efflux is unlikely to be the reason why CBD is inactive against most Gram-negative bacteria. CBD was marginally activity against an *E. coli lpxC* cell membrane mutant (MIC 128 µg mL$^{-1}$), suggesting that the

**Table 1 Minimum inhibitory concentrations (MICs) of CBD (µg mL⁻¹).**

**(a) Gram-positive (resistant strains in bold)**

| Species | Strain | Vancomycin | Daptomycin | Trimethoprim | Mupirocin | Clindamycin | CBD |
|---|---|---|---|---|---|---|---|
| Staphylococcus aureus | ATCC 25923 MSSA | 1–2 | 1–2 | 2–4 | 0.25–0.5 | 0.06–0.25 | 1–2 |
| **Staphylococcus aureus** | ATCC 43300 MRSA | 0.5–1 | 0.5–1 | 2 | 0.125–0.5 | >64 | 1–2 |
| **Staphylococcus aureus** | NRS-1 MRSA VISA | 2–4 | 4–8 | 2–4 | 0.125–0.5 | 64 | 1–4 |
| **Staphylococcus aureus** | VRS1 VRSA | >64 | 1–4 | >64 | 32–64 | >64 | 1–2 |
| Staphylococcus epidermidis | ATCC 12228 | 1–2 | 1–8 | 1–4 | 0.25–0.5 | 0.06–0.25 | 1–2 |
| **Staphylococcus epidermidis** | NRS-60 VISE | 2–4 | 1–4 | >64 | 0.125–0.5 | 0.06–0.125 | 4–8 |
| Streptococcus pneumoniae | ATCC 33400 | 0.5–1 | 1–4 | 0.5–2 | 0.25–0.5 | 0.06–2 | 1–2 |
| **Streptococcus pneumoniae** | ATCC 700677 MDR | 1 | 1–4 | 1–2 | 0.25–0.5 | >64 | 1–4 |
| Streptococcus pyogenes | ATCC 12344 | 0.25–0.5 | 0.125–0.25 | 2–4 | ≤0.03 | 0.125–0.25 | 1 |
| Enterococcus faecium | ATCC 35667 | 0.25–0.5 | 4–8 | ≤0.03–0.06 | 1 | ≤0.03–0.06 | 0.5–1 |
| **Enterococcus faecium** | ATCC 700221 | >64 | 8 | 16 | 2 | >64 | 1 |
| Enterococcus faecalis | ATCC 29212 | 1–2 | 4–8 | 0.25–0.5 | 64 | 16–32 | 2 |
| **Enterococcus faecalis** | clinical isolate | >64 | 8 | >64 | >64 | >64 | 2–4 |

| Species | Strain | Vancomycin | | Levofloxacin | Meropenem | Gentamicin | CBD |
|---|---|---|---|---|---|---|---|
| **Enterococcus faecalis** | MMX 486 VRE | >16 | | >32 | 4 | >32 | 1 |
| Enterococcus faecium | ATCC 19434 | 0.25 | | 4 | 8 | 4 | 0.5 |
| **Enterococcus faecium** | MMX 485 VRE | >32 | | 1 | >32 | >32 | 1 |

Anaerobic growth conditions

| Species | Strain | Vancomycin | Erythromycin | Tetracycline | Mupirocin | Clindamycin | CBD |
|---|---|---|---|---|---|---|---|
| **Staphylococcus aureus** | ATCC 43300 MRSA | 0.5–1 | >32 | 0.06–0.25 | 0.03–0.06 | >32 | 1–2 |
| Cutibacterium acnes | ATCC 6919 | 0.25 | 0.125–0.25 | 0.125–0.5 | >32 | 0.125 | 1–2 |
| Clostridioides difficile | M7404 human ribotype 027 | | | | | | 2–4 |

**(b) Gram-negative**

| Species | Strain | Colistin | Trimethoprim | Mupirocin | Clindamycin | CBD |
|---|---|---|---|---|---|---|
| Escherichia coli | ATCC 25922 | 0.06–0.125 | 0.5–2 | >64 | >64 | >64 |
| **Klebsiella pneumoniae** | ATCC 700603, ESBL | 0.125–1 | 8–16 | >64 | >64 | >64 |
| Pseudomonas aeruginosa | ATCC 27853 | 0.25–1 | >64 | >64 | >64 | >64 |

| Species | Strain | Vancomycin | Levofloxacin | Meropenem | Gentamicin | CBD |
|---|---|---|---|---|---|---|
| Acinetobacter baumannii | ATCC 19606 | >32 | 0.25 | 0.5 | 8 | >64 |
| Serratia marcescens | MMX 6462 | >32 | 0.06 | 4 | 1 | >64 |
| Stenotrophomonas maltophila | MMX 4746 | >32 | 0.5 | 8 | 0.5 | >64 |
| Burkholderia cepacia | MMX 547 | >32 | 1 | 1 | >32 | >64 |
| Proteus mirabilis | MMX 6442 | >32 | 0.03 | ≤0.03 | 0.25 | >64 |
| Salmonella typhimurium | ATCC 35987 | >32 | 0.03 | ≤0.03 | 0.25 | >64 |
| Shigella dysenteriaev | ATCC 29026 | >32 | 0.03 | ≤0.03 | 0.5 | >64 |
| Haemophilus influenzae | ATCC 49247 | >32 | 0.015 | 0.5 | 1 | >64 |
| Moraxella catarrhalis | MMX 3782 | 32 | 0.03 | ≤0.03 | 0.25 | 1 |
| Neisseria gonorrhoeae | ATCC 49226 | 8 | 0.002 | ≤0.03 | 4 | 1 |
| Neisseria meningitidis | ATCC 13090 | >32 | 0.004 | ≤0.03 | 8 | 0.25 |
| Legionella pneumophila | MMX 7515 | 0.12 | 0.5 | 0.25 | 1 | 1 |

VISA vancomycin intermediate S. aureus, VISE vancomycin intermediate S. epidermidis.

altered lipopolysaccharide architecture (i.e., lack of lipid A) could allow CBD penetration into the cells (Supplementary Table 9). Similarly, against a lipid A deficient A. baumanni strain[30], where the MIC of polymyxin increases from 0.25 to >128 µg mL⁻¹, the MIC CBD decreased from >128 to 0.25 µg mL⁻¹ (Supplementary Table 9) with a similar increase in susceptibility seen for teicoplanin and gentamicin (as reported by Moffatt et al.[30]). This increased activity in the presence of altered membrane structure is further supported by synergy studies with the membrane-disrupting antibiotics polymyxin B and colistin. Although CBD was generally inactive (>64 µg mL⁻¹) against the ESKAPE Gram-negative bacteria when tested alone, in some cases it was found to synergize (analyzed by FICI calculation) with colistin and polymyxin B: e.g., against E. coli and A. baumannii, but not against K. pneumoniae or P. aeruginosa. For example, sub-MIC concentrations of colistin or polymyxin B (0.06 and 0.25 µg mL⁻¹,

respectively) reduced to the MIC of CBD to 4–8 µg mL⁻¹ against A. baumannii ATCC 19606. Similarly, a sub-MIC concentration of colistin or polymyxin (0.125 µg mL⁻¹) saw CBD active at 4–16 µg mL⁻¹ against E. coli ATCC 25922 (Supplementary Table 10). The same effect was observed with polymyxin B nonapeptide (PMBN), a membrane-disrupting polymyxin analog without inherent antibacterial activity[31]. While PMBN was inactive against the four species (MIC >32 µg mL⁻¹), as little as 2–4 µg mL⁻¹ of PMBN was able to reduce the CBD MIC from >256 µg mL⁻¹ to 32–64 µg mL⁻¹ against E. coli, P. aeruginosa, and A. baumannii, though not K. pneumoniae (Supplementary Table 10). Synergy was not observed with other classes of antibiotics, including glycopeptides (teicoplanin), various types of β-lactams (ceftazidime, cefepime aztreonam, meropenem), and a different type of membrane-active peptide antibiotic, the β-hairpin antimicrobial peptide arenicin-3.

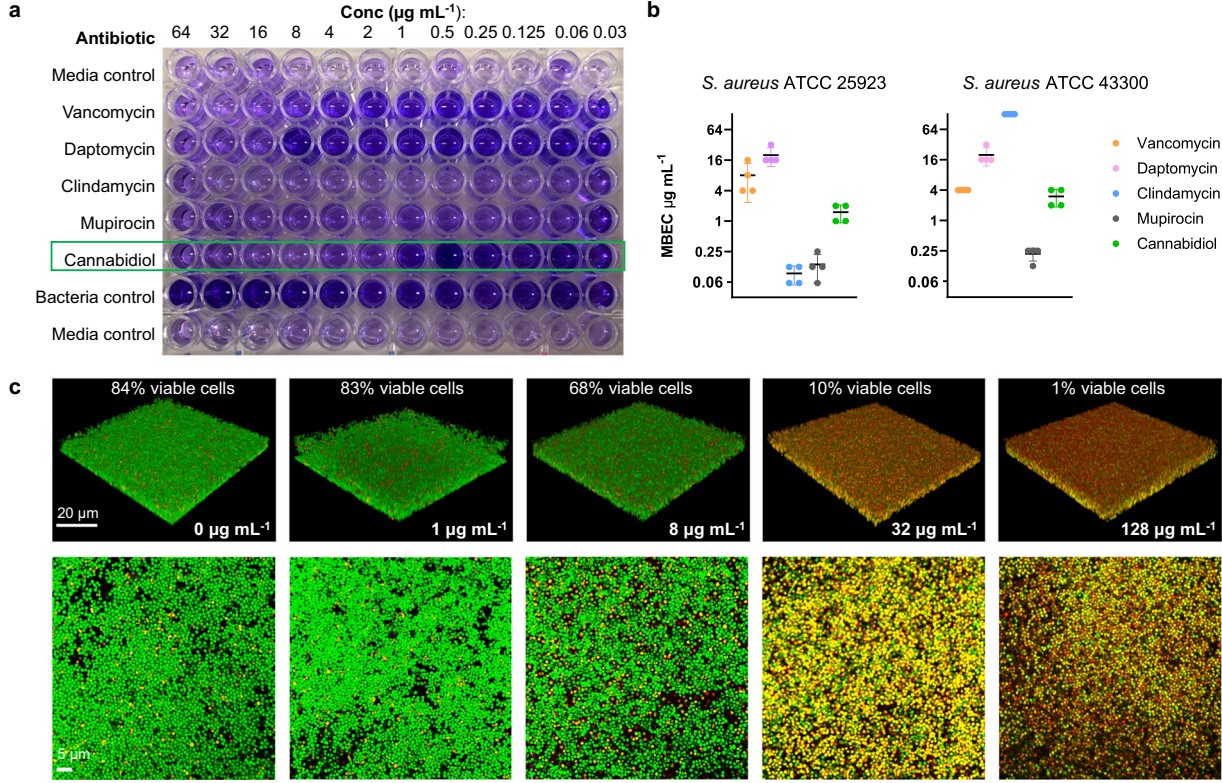

**Fig. 2 Anti-biofilm activity of cannabidiol.** Minimum biofilm eradication concentration (MBEC) assessment of CBD using crystal violet staining (**a**) to assess (**b**) MSSA ATCC 25923 and MRSA ATCC 43300 biofilm remaining after 24 h incubation with CBD, vancomycin, daptomycin, trimethoprim, mupirocin, or clindamycin (biofilm initially established by 48 h growth in TSB + 5% glucose). Data are $n = 4$ biologically independent samples. **c** Confocal microscopy of MRSA ATCC 43300 biofilm grown on microscope slides, then treated with CBD at varying concentrations for 24 h. Slides were then stained with SYTO 9 (green, indicates live + dead cells) and propidium iodide (red, indicates dead cells) nucleic acid stains (white scale bars represent 20 μm in the 3D panels and 5 μm in the 2D panels).

Additional microbiological characterization studies focused on *S. aureus* biofilms, as the ability to inhibit biofilm formation or eradicate established biofilms should improve the ability of an antibiotic to treat infections. CBD was active against both MSSA and MRSA biofilms, with a minimum biofilm eradication concentration (MBEC) of 1–2 and 2–4 μg mL$^{-1}$ respectively, similar to its MIC. The MBEC values were substantially better than daptomycin or vancomycin against MSSA, and daptomycin and clindamycin against MRSA (Fig. 2a, b). Confocal microscopy of CBD-treated biofilms showed that CBD was able to penetrate and kill the biofilm (Fig. 2c), as assessed by staining with green SYTO 9 dye (membrane penetrable dye with high affinity for DNA and stains both live and dead bacteria) and red propidium iodide (PI) dye (stains nuclear chromatin upon cell membrane disruption, resulting in fluorescence enhancement)[32,33]. Higher concentrations of CBD were needed to see >90% killing (32 μg mL$^{-1}$) than the MBEC measured using the crystal violet assay (2–4 μg mL$^{-1}$), presumably due to the differences in assay conditions and readouts.

A critical parameter that must be assessed for any potential new antibiotic is the propensity for resistance to emerge. CBD showed a very low innate resistance frequency value (<3.78 × 10$^{-10}$ at 2 × MIC vs. MRSA ATCC 43300) (Supplementary Table 11). More importantly, CBD demonstrated a low propensity to induce resistance against MRSA ATCC 43300 (1.5-fold increase in MIC over 20 days of daily passage compared to 26-fold increase for daptomycin: average of 8 replicates each) (Fig. 3). The increase in MIC in the daptomycin samples varied considerably between replicates, presumably due to the stochastic nature of mutation events leading to high levels of resistance.

Similarly, three strains of *C. acne* were grown for 15 passages (less passages due to slow growth requiring 48 h for each passage) in the presence of CBD, clindamycin and erythromycin. In this case, none of the antibiotics induced resistance (Supplementary Fig. 1).

Finally, CBD was tested for toxicity against mammalian cells, with no signs of hemolysis up to 256 μg mL$^{-1}$ when exposed to human red blood cells (Supplementary Table 12). Modest cytotoxicity was seen against HEK-293 (human embryonic kidney) cells, with a CC$_{50}$ around 200 μg mL$^{-1}$ (Supplementary Table 13); a previous report of cytotoxicity against a range of cancer cell lines indicated 50% cell viability seen at 20 μM CBD concentration[34].

**Mechanism of action of cannabidiol.** Radiolabeled macromolecular synthesis assays in *S. aureus* RN42200 showed that protein, DNA, RNA and peptidoglycan synthesis were all sharply inhibited at concentrations near the MIC (2–3 μg mL$^{-1}$), consistent with a rapid bactericidal action shutting down all these synthesis pathways (Fig. 4a, control antibiotics shown in Supplementary Fig. 2). Only lipid synthesis showed signs of reduction at concentrations below the MIC, supporting prior speculation of membrane-based effects[23]. Additional evidence for membrane activity is provided by a concentration-dependent membrane depolarization seen in MRSA (Fig. 4b), but not *E. coli* (Fig. 4c), measured using the membrane potential-sensitive cationic fluorescent probe 3,3′-dipropylthiadicarbocyanine iodide [DiSC3 (5)]. This dye is prevented from partitioning to the surface of polarized cells during disruption of membrane potential, causing dye release into the media that results in a fluorescent signal[35].

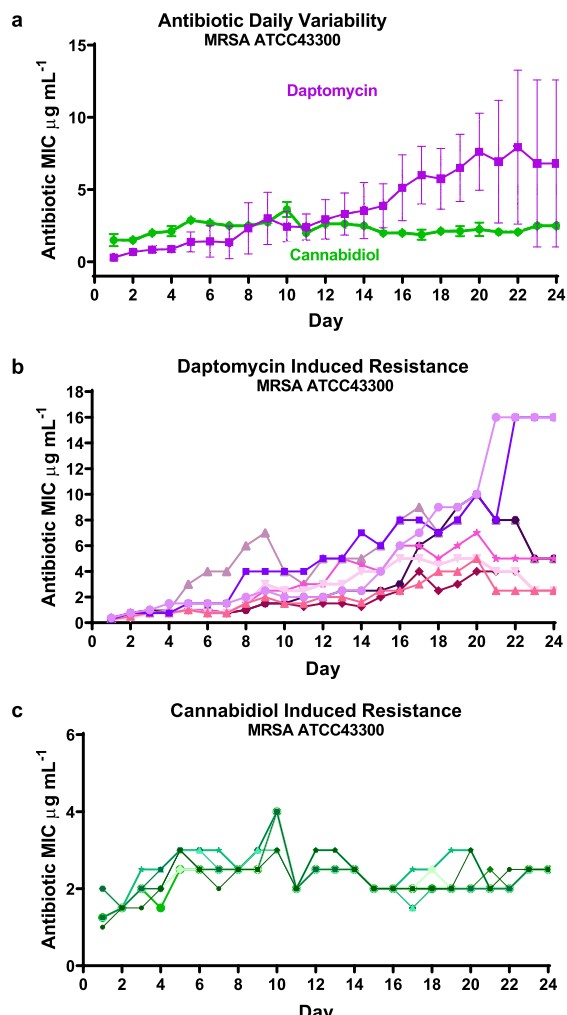

**Fig. 3 Broth dilution serial passage resistance induction studies.**
**a** Average daily MIC during exposure of MRSA (ATCC 43300) to sublethal concentrations of daptomycin or CBD over 20 days of bacterial growth, followed by 5 days without antibiotic exposure. Data are mean ± SEM for $n = 8$ biologically independent samples. **b, c** The corresponding individual replicates for the two compounds (drug-free from Day 21).

Finally, bacterial cytological profiling (BCP) gave results (Fig. 4d–g) consistent with previously published BCP results for other antibiotics known to act by membrane permeabilization[36–38]. Fluorescent microscopy showed uptake of SYTOX™ Green dye (normally not cell-penetrable)[38] in a concentration-dependent and time-dependent manner, in as little as 10 min at 1× MIC for both *S. aureus* ATCC 29213 (Fig. 4d, f) and *B. subtilis* PY79 (Fig. 4e, g). Taken together, all of these results are consistent with CBD acting very rapidly to disrupt bacterial cytoplasmic membranes, though whether a specific molecular target is involved remains to be determined.

**Efficacy of cannabidiol**. As discussed earlier, the initial interest in examining CBD's antimicrobial activity was driven by the use of topical CBD formulations in clinical programs for the treatment of acne and atopic dermatitis. We initiated efficacy studies focused on screening for antibacterial activity in an ex vivo porcine *S. aureus* skin infection model[39]. CBD formulations of varying composition and consistency, including liquids, gels, creams and ointments, were evaluated for their ability to kill

MRSA inoculated on pig skin, testing the killing effectiveness at both 1 h and 24 h following application (Fig. 5a, b). CBD concentrations ranged from 5 to 20% in formulations based on different silicones (formulations #4–12), petrolatum (mineral oil jelly, #1), transcutol (diethylene glycol monoethyl ether, #2) and polyethylene glycol (PEG 4000/400, #3) (Supplementary Table 14). Efficacy was highly formulation-dependent, and some formulation vehicles had modest to good antimicrobial activity on their own (e.g., liquid formulation #2, with a high content of transcutol and 3.4% isopropyl alcohol). Good activity (2-log to 3-log reduction in colony-forming units [CFU] after 1 h, >5 log reduction at 24 h) was consistently observed with Formulations #3 and #12 but not their corresponding vehicles. Formulation #3 is a PEG-based ointment formulation with 20% CBD, with the base matching the formulation used for Bactroban/mupirocin ointment. Formulation #12 is a gel formed from a mixture of a silicone fluid (polydimethylsiloxane liquid) and transcutol combined with a gelling agent (Dow Corning BY 11-030) and a small amount of water, again with 20% CBD. Formulations #3 and #12 were then tested for concentration dependence with 5, 10, 15, and 20% CBD content (Fig. 5c, d). While some concentration effects were evident, particularly after only 1 h, by 24 h all caused significant reductions in bacterial load ($p < 0.001$). The 5, 10 and 15% formulations were not statistically different from the 20% formulations; though there were trends for reduced activity with the 5% formulations. Finally, to ensure that efficacy was maintained against clinical isolates, 20% CBD formulations of #2, #3, and #12 were tested against other MRSA strains with low or high levels of mupirocin resistance ((Fig. 5e–h). A 2% mupirocin formulation (used clinically for nasal decolonization) was still active against the low-level resistant isolate (MIC mupirocin 16–32 μg mL$^{-1}$, CBD 3 μg mL$^{-1}$) but lost effectiveness against the highly-resistant isolates (MIC mupirocin 256–2048 μg mL$^{-1}$, CBD 1.5–3 μg mL$^{-1}$). Although the CBD MIC was not substantially different against the high-level mupirocin-resistant strains, the effectiveness of the CBD formulations at reducing the bacterial load was attenuated. Notably, the vehicle used in Formulation #2 again showed substantial activity against the ATCC 43300 isolate, as well as the low level mupirocin-resistant isolate, but was also much less active against the high-level mupirocin-resistant strains.

In parallel with the formulation experiments, 5% CBD in a non-optimized volatile silicone-based vehicle (predominantly low viscosity polydimethylsiloxane liquid) was tested in a mouse topical infection model, in which a bioluminescent *S. aureus* (Xen-29) infection was established in disrupted skin of immunocompromised mice. This study showed CBD led to statistically significant reductions in *S. aureus* at 48 h compared to vehicle ($p = 0.0184$), though the CBD formulation was not as effective as the 2% mupirocin positive control (I = <0.0001 for mupirocin vs. vehicle) (Fig. 5i–k). We also evaluated CBD's systemic antimicrobial activity in an immunocompromised thigh infection mouse model against MRSA ATCC 43300. However, despite the successful systemic treatment of other types of diseases with CBD, it was ineffective when systemically dosed subcutaneously at 100 mg kg$^{-1}$, intraperitoneally at 200 mg kg$^{-1}$, or orally at 250 mg kg$^{-1}$ (Supplementary Fig. 3).

**Structure-activity relationships of cannabidiol analogs**. We initiated a medicinal chemistry campaign to attempt to improve the properties of CBD so that it can be used systemically to treat infections, with a focus on reducing protein binding/serum reversal while retaining or improving antibacterial activity. We have tested both naturally occurring cannabinoids along with synthetic analogs designed to examine the importance of key structural features

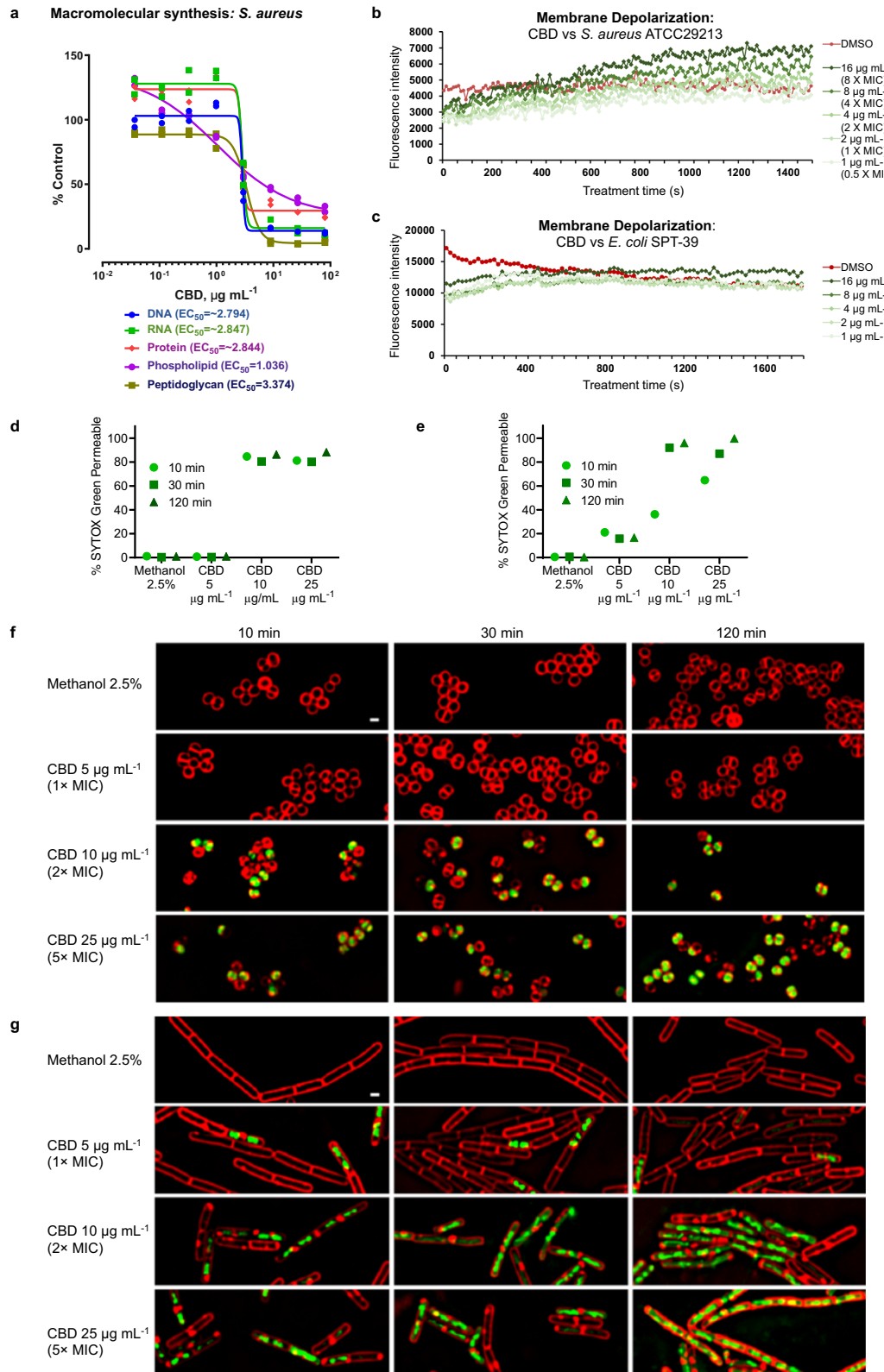

(Table 2). The chemistry of cannabinoids has recently been reviewed[40]. There were clear differences in the structure-activity relationship (SAR) for the Gram-positive activity against MRSA compared to the Gram-negative activity against *N. gonorrhoeae*. For example, oxidation of the cyclohexene methyl group to a hydroxymethyl (7-hydroxy-cannabidiol [7-OH-CBD] **2**) or carboxyl

group (7-nor-7-carboxy-cannabidiol [7-COOH-CBD] **4**) was detrimental to MRSA activity (potency decreased from 1–2 to 16 and >64 μg mL$^{-1}$, respectively), but had little impact on *N. gonorrhoeae* activity (MIC between 0.125–2 μg mL$^{-1}$). In contrast, positioning a carboxyl substituent on the resorcinol aromatic ring had less effect on MRSA potency, with an MRSA MIC of 16–32 μg mL$^{-1}$ for

**Fig. 4 Mode of action of cannabidiol. a** Macromolecular synthesis assay in *S. aureus* RN42200 showing inhibition of radiolabelled substrate uptake in DNA ([2-$^{14}$C]-thymidine), RNA ([5,6-$^{3}$H]-uracil), protein (L-[4,5-$^{3}$H]-leucine), phospholipid ([2-$^{3}$H]-glycerol) and peptidoglycan ([$^{14}$C(U)]-glycine) synthesis pathways after 35 min incubation. Data are mean ± SD for $n = 2$ biologically independent samples. MIC for CBD is 2–3 μg mL$^{-1}$. **b**, **c** Membrane depolarization assay in *S. aureus* ATCC 29213 **b** and *E. coli* SPT-39 **c** monitoring uptake of potential-sensitive fluorescent dye 3,3-dipropylthiadicarbocyanine iodide [DiSC3(5)] over time in presence of increasing concentrations of CBD. **d–g** Bacterial cytological profiling (BCP) assay in *S. aureus* ATCC 29213 **d**, **f** or *B. subtilis* PY79 **e**, **g** showing uptake of SYTOX™ Green dye over time in the presence of increased concentrations of CBD. Red FM 4–64 dye is used to visualize membranes (white scale bar is 1 μm). The plots **d**, **e** quantify the percentage of cells that have been permeabilized, defined as the fraction of cells with a mean SYTOX™ Green intensity greater than a cut-off value of 250 (number of cells measured ranged from 945–8050 for each time/concentration). Since CBD was dissolved in methanol, control cells were treated with 2.5% methanol.

cannabidiolic acid (CBDA **14**). In the context of CBG or THC backbones, an aromatic carboxyl group had little effect for either bacteria (e.g., tetrahydrocannabivarin [THCV] **19** vs. tetra-hydrocannabivarinic acid [THCVA] **20**, CBG **25** vs. cannabigerolic acid [CBGA] **26**). The extra ring cyclisation in the (−)-Δ$^8$-THC **23** backbone led to a 2-fold to 4-fold reduction in activity for both MRSA and *N. gonorrhoeae*, while the acyclic alkyl substituents found in CBG **25** and CBGA **26** reduced MRSA activity 2-fold to 4-fold compared to CBD, but were within the range of CBD *N. gonorrhoeae* activity.

Extending the CBD aromatic side chain substituent from a pentyl (**1**) to heptyl (**9**) chain gave similar or marginally improved potency for MRSA (from 1–2 μg mL$^{-1}$ to 0.5–2 μg mL$^{-1}$, within the assay range of variability) but reduced it for *N. gonorrhoeae* (0.125–2 μg mL$^{-1}$ to 0.5–4 μg mL$^{-1}$). Shortening to a propyl group (cannabidivarin [CBDV] **6**) slightly decreased MRSA activity (2–4 μg mL$^{-1}$) but improved *N. gonorrhoeae* activity (≤0.03–0.5 μg mL$^{-1}$). Unfortunately, the serum reversal effect for MRSA was not improved by the reduction in length of the lipophilic chain (MIC >256 μg mL$^{-1}$). A phenylbutyl substituent (**10**) was equivalent to the *n*-pentyl group of CBD for both bacteria. The site of attachment of the cyclohexene substituent was not critical—when attached between one of the phenol groups and the alkyl substituent instead of between the two phenolic groups (e.g., scaffold **B** vs. **A**, compounds **15** and **16**), activity was only reduced 2-fold or remained the same.

The phenol groups were important: when both were methylated (**13**) activity was lost against both bacteria (>64 μg mL$^{-1}$). The removal of one phenol group (**11**) blocked MRSA activity but *N. gonorrhoeae* activity was largely maintained. Mono-methylation (**12**) also abolished MRSA activity and substantially reduced *N. gonorrhoeae* inhibition. The alkene bonds present in the isopropene-substituted cyclohexenyl substituent were not important: reduction of both (**17**) improved both MRSA and *N. gonorrhoeae* activity, while reduction of just the isoprene group (**18**) had little effect.

Finally, we tested the potential to conduct modifications at the 7-methyl position by preparing a methyl amide derivative (**5**) of 7-COOH-CBD (**4**) and an *O*-methyl derivative (**3**) of 7-OH-CBD (**2**): both were less active than CBD. The *O*-methyl derivative **3** had the same activity as the hydroxyl parent **2** for MRSA (16 μg mL$^{-1}$) but was substantially less active for *N. gonorrhoeae* (4–16 vs. 0.25–2 μg mL$^{-1}$). The methyl amide **5** was more active than the acid parent **4** against MRSA (16 vs. >64 μg mL$^{-1}$) but slightly less active for *N. gonorrhoeae* (2–4 vs. 0.5–2 μg mL$^{-1}$).

## Discussion

In this study we have discovered that CBD and other cannabinoids have selective activity against a subset of Gram-negative bacteria that includes the CDC urgent priority and World Health Organisation (WHO) high priority drug-resistant pathogen *N. gonorrhoeae*. This bacteria causes infections that are increasingly unresponsive to existing antibiotics. We also describe a much

more extensive spectrum of Gram-positive activity than previously reported, including against the WHO high priority pathogens vancomycin-resistant *E. faecium* and MRSA, and a large set of clinical isolates. Importantly, we also demonstrate that CBD does not lead to resistance after repeated exposure. This combination of properties provides a compelling case to conduct further investigations into this underexplored class of compounds.

The Achilles heel for further advancement of CBD as an effective therapeutic antibiotic is its lack of systemic activity. This liability can be explained by CBD's very high (up to >94%) serum binding[28], and it is largely inactivated when MIC assays are conducted in the presence of 50% human serum (MRSA MIC > 256 μg mL$^{-1}$). Published pharmacokinetic studies of oral, subcutaneous, pulmonary and intravenous dosing of CBD in mice[41,42], rat[42–44], dog[45], monkey[46], and human[47,48] indicate that the concentrations administered in our thigh infection studies should provide total plasma concentrations that exceed the MIC for a number of hours, but not for the free CBD fraction if it is highly plasma bound and therefore unavailable. Interestingly, these reports indicate that oral delivery appears to provide much better systemic exposure than subcutaneous dosing[43]. What is more surprising from the current studies is the high dependence on formulation composition for topical efficacy in an ex-vivo pig skin infection model, as high levels of CBD are not able to kill bacteria unless delivered in a compatible vehicle.

Our initial SAR investigations provide evidence of the potential to alter CBD's core structure while retaining activity, giving hope that the physicochemical properties can be modified to provide systemic activity. Notably, variations were seen in the SAR against MRSA and *N. gonorrhoeae*, meaning there is potential to develop a *N. gonorrhoeae* targeting agent that is not only selective over other Gram-negative bacteria, but also over Gram-positive bacteria. Narrow spectrum antibiotics are increasingly recognized as an important advancement in antibiotic technology, with the ability to spare the natural microbiome by not killing beneficial commensal bacteria[49]. This could potentially provide a substantial advantage for cannabidiol derivatives over other classes of antimicrobial compounds. Our experiments demonstrate that the lack of CBD activity against most Gram-negative bacteria is related to the presence of the outer membrane and lipopolysaccharide, as membrane-disrupting drugs or LPS-deficient bacteria increase Gram-negative susceptibility to CBD. Curiously, CBD-susceptible *N. gonorrhoeae* and *Moraxella catarrhalis* are both Gram-negative species where lipopolysaccharide is not an essential outer membrane building block, but then so is non-susceptible *A. baumannii*[50]. Clearly, more work is required to decipher the subtle variations in outer membrane composition that affect CBD antibacterial activity, as well as further studies to ascertain whether there are specific pathways targeted by CBD.

In summary, CBD represents the prototypical member of an exciting structural class of antibiotics, with potential to develop new analogs that have narrow spectrum selective Gram-negative activity against the dangerous pathogen *N. gonorrhoeae*. Given recent concerns over the rise of 'super gonorrhea'[51], the discovery

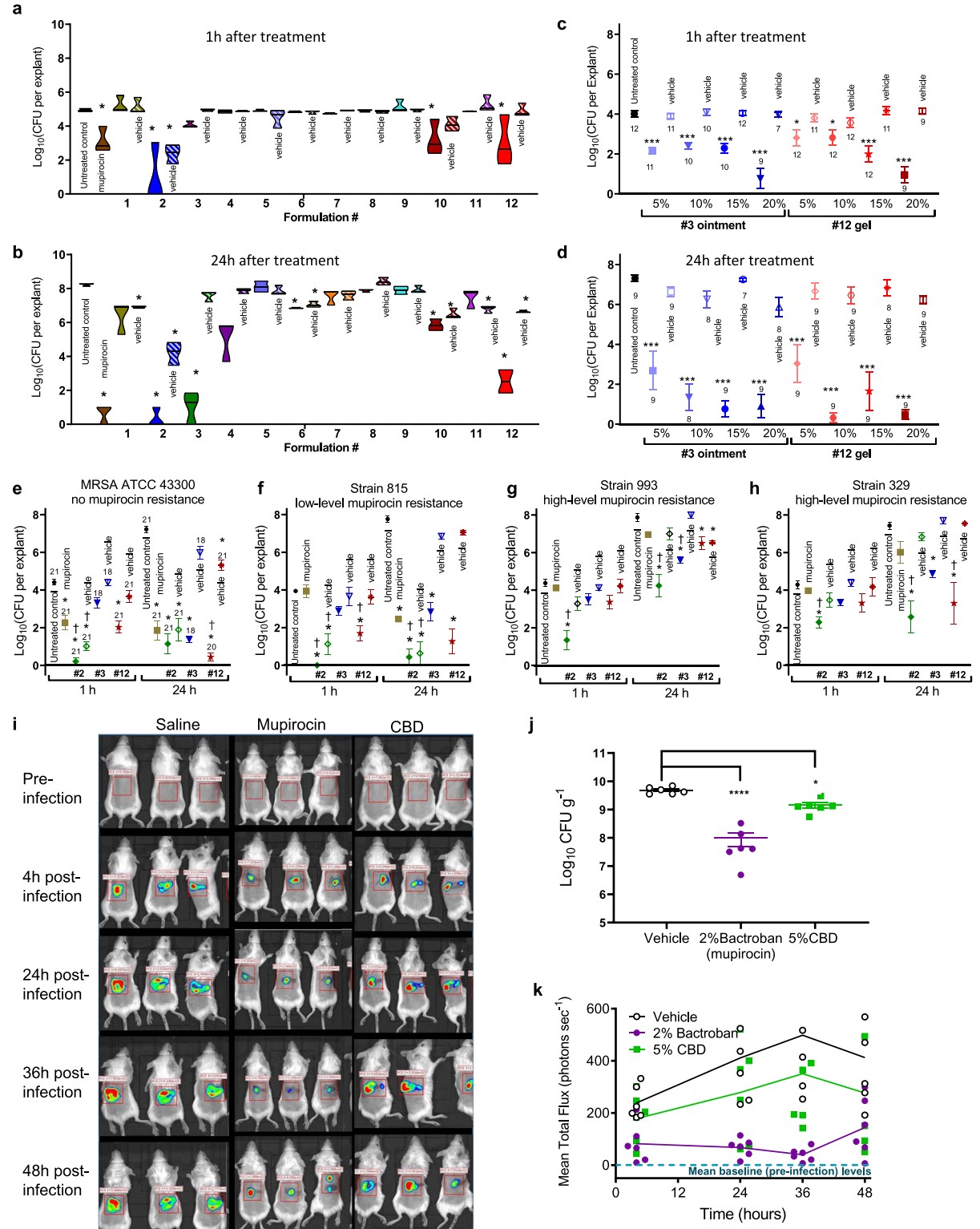

of a new class of compounds that is active against the major types of *N. gonorrhoeae* resistance is an important development, though tempered by the need to develop an analog suitable for systemic treatment. For a more immediate application of CBD's antimicrobial activity, clinical trials are being initiated to test CBD as a topical therapy for nasal decolonization of *S. aureus*[52], with the eventual goal of reducing *S. aureus* infections during surgery. The safety of CBD (given its widespread existing use) and its low propensity to induce resistance make this indication an attractive

target, and the FDA has recently granted Qualified Infectious Disease Product (QIDP) status[53].

## Methods

**Ethics approvals**. Research was conducted under an Institutional Animal Care and Use Committee approved protocol in compliance with the Animal Welfare Act, PHS Policy and other federal statutes and regulations relating to animals and experiments involving animals. Contract facilities where animal studies were conducted were accredited by the Association for Assessment and Accreditation of

**Fig. 5 Efficacy studies of cannabidiol. a**, **b** Ex vivo pig skin model. Colony forming units (CFU) remaining on 5 mm biopsy pig skin explants inoculated with $2 \pm 0.5 \mu L$ of ~$5 \times 10^8$ CFU mL$^{-1}$ MRSA ATCC 43300 and incubated at 37 °C in 6-well plates containing a 0.4 µm trans-well insert. Formulations (see Supplementary Table 10) containing CBD or mupirocin (solid colors) and vehicle formulations with no CBD (barred colors) were applied (150 µL) 2 h post-infection. At 1 h (**a**) or 24 h (**b**) later tissue was removed and the remaining CFU quantified: $n = 3$ independent experiments except for formulation 8 at 1 h and formulation 12 at 12 h, $n = 2$ independent experiments, error bars show SEM; statistical analysis done using GraphPad Prism 8, 2-way ANOVA, Dunnett's multiple comparisons test, asterisk (*) denotes statistically significant deviation from Growth Control ($p < 0.05$). **c**, **d** Concentration-dependence of ointment #3 and gel #12 formulations containing varying % of CBD (solid symbols) and corresponding vehicle only (open symbols) against MRSA ATCC 43300 in same pig skin model: $n = 7$–12 biologically independent samples as indicated near each data symbol, error bars show SEM; statistical analysis done using GraphPad Prism 8, one-way ANOVA with Dunnet post-correction, asterisk (*) denotes statistically significant deviation from Growth Control ($p < 0.05$), **($p < 0.01$), ***($p < 0.001$)). **e**–**h** Effectiveness of 2% mupirocin and 20% CBD formulations (liquid #2, ointment #3, and gel #12) (solid symbols) and corresponding vehicle only (open symbols) against mupirocin sensitive and resistant MRSA strains in same pig skin mode: $n = 18$–21 biologically independent samples for **e** (indicated near each data symbol), $n = 3$ for **f**, $n = 6$ for **g**, $n = 6$ for **h**, error bars show SEM; statistical analysis done using GraphPad Prism 8, 2-way ANOVA, Dunnett's multiple comparisons test, asterisk (*) denotes statistically significant deviation from Growth Control, and † indicates significant deviation from mupirocin-treated skin (both $p < 0.05$)). **i**–**k** In vivo topical skin infection model. Immunocompromised mice ($n = 6$ group) were shaved on their back and the skin surface disrupted, then inoculated with $5 \times 10^7$ CFU of Xen-29 S. aureus bacteria in a 10 µL droplet. Treatment was initiated immediately after inoculation and repeated at 12, 24, and 32 h after infection, with 50 µL of vehicle, 2% mupirocin, or 5% CBD. Animals were imaged using the Lumina II system (Perkin-Elmer) at 4, 24, 36, and 48 h post-infection (**i**, only 3 mice shown per image) and bioluminescence (photons per second) determined (**j**). At 48 h after the first treatment and infection, animals were sacrificed and samples of skin from each animal were homogenized and plated on nutrient agar in order to determine the CFU's per animal (**k**). Errors are mean ± SEM, $n = 6$ animals. Statistical analysis done using GraphPad Prism 8, 1-way ANOVA, Bonferroni post-test.

**Table 2 Structure of CBD analogs and summary of effect of modification to different positions on antimicrobial activity against MRSA and *N. gonnorrhoeae* (MIC, µg mL$^{-1}$, $n = 4$).**

| # | Compound name | Core Scaffold | R1 | R2 | R3 | R4 | R5 | S. aureus MRSA ATCC 43300 | N. gonorrhoeae ATCC 19424 |
|---|---|---|---|---|---|---|---|---|---|
| | Control: Vancomycin | | | | | | | 1 | nd |
| | Control: Colistin | | | | | | | nd | 32 |
| | Control: Daptomycin | | | | | | | 1–2 | >64 |
| | Control: Clindamycin | | | | | | | >64 | 1 |
| | Control: Mupirocin | | | | | | | 0.125–0.5 | ≤0.03–0.06 |
| 1 | Cannabidiol | A | nPr | H | OH | OH | Me | 1–2 | 0.125–2 |
| 2 | 7-hydroxycannabidiol | A | nPr | H | OH | OH | CH$_2$OH | 16 | 0.25–2 |
| 3 | MTC-005 | A | nPr | H | OH | OH | CH$_2$OMe | 16 | 4–16 |
| 4 | 7-nor-7-carboxycannabidiol | A | nPr | H | OH | OH | CO$_2$H | 64->64 | 0.5–2 |
| 5 | MTC-002 | A | nPr | H | OH | OH | CONHMe | 16 | 2–4 |
| 6 | Cannabidivarin | A | Me | H | OH | OH | Me | 2–4 | ≤0.03–0.5 |
| 7 | 7-nor-7-hydroxymethyl-cannabidivarin | A | Me | H | OH | OH | CH$_2$OH | 64 | >64 |
| 8 | 7-nor-7-carboxy-cannabidivarin | A | Me | H | OH | OH | CO$_2$H | >64 | 2–16 |
| 9 | MTC-007 | A | nPent | H | OH | OH | Me | 0.5–2 | 0.5–4 |
| 10 | MTC-017 | A | (CH$_2$)$_2$Ph | H | OH | OH | Me | 2 | 0.5–4 |
| 11 | MTC-011 | A | nPr | H | H | OH | Me | 64->64 | 1–8 |
| 12 | MTC-012 | A | nPr | H | OMe | OH | Me | >64 | 8–32 |
| 13 | MTC-013 | A | nPr | H | OMe | OMe | Me | >64 | >64 |
| 14 | Cannabidiolic acid (CBDA) | A | nPr | CO$_2$H | OH | OH | Me | 16–32 | 1–2 |
| 15 | MTC-018 | B | nPent | – | – | – | – | 1–2 | 1–2 |
| 16 | MTC-008 | B | (CH$_2$)$_2$Ph | – | – | – | – | 2–4 | 1–4 |
| 17 | MTC-014 | C | – | – | – | – | – | 0.25–0.5 | 0.25–0.5 |
| 18 | MTC-009 | D | – | – | – | – | – | 1–2 | 0.5–2 |
| 19 | Tetrahydrocannabivarin (THCV) | E | Me | H | – | – | – | 64 | 16 |
| 20 | Tetrahydrocannabivarinic acid (THCVA) | E | Me | CO$_2$H | – | – | – | 32–64 | 4–16 |
| 21 | (−)-Δ$^9$-THC | E | nPr | H | – | – | – | 4–8 | 4–8 |
| 22 | Δ$^9$-tetrahydrocannabinolic acid A (THCA-A) | E | nPr | CO$_2$H | – | – | – | 8 | 0.5–8 |
| 23 | (−)-Δ$^8$-THC | F | – | – | – | – | – | 4–8 | 2–4 |
| 24 | Cannabinolic acid A (CBNA) | G | – | – | – | – | – | 4–16 | 0.25–4 |
| 25 | Cannabigerol (CBG) | H | – | H | – | – | – | 4–8 | 1–2 |
| 26 | Cannabigerolic acid (CBGA) | H | – | CO$_2$H | – | – | – | 2–4 | 1–2 |

Laboratory Animal Care, International and adhere to principles stated in the Guide for the Care and Use of Laboratory Animals, National Research Council, 2011. Studies performed at the University of Queensland were approved by the Molecular Biosciences Animal Ethics Committee. Sample size for in vivo studies was selected based on minimizing animal use while providing sufficient data points to show significant differences based on historical studies. No animals were excluded from analysis. No documented method of randomization was employed, with animals randomly assigned to groups. Code numbers were employed for all in vivo studies, so the study investigator was blinded to the actual identity of compounds. Human ethics approval from the University of Queensland Institutional Human Research Ethics Committee was obtained for use of human blood for hemolysis studies (approval number 201400003).

### Chemical synthesis

*General.* Cannabidiol **1** was obtained from AMRI (Batch R0030516 RM342K.0706). Natural analogs **2**, **4**, **14**, and **19–26** were purchased from Kinesis Australia Pty Ltd., the Australian distributor for Cerilliant Corporation. Synthetic analogs **3**, **5–13**, and **15–18** were synthesized by BDG Synthesis using modified literature methods, as described below. Compound purity was determined using HPLC with UV detection, with structures confirmed by HRMS and $^{1}$H NMR (500 MHz, Bruker) and $^{13}$C NMR (125 MHz, Bruker) experiments. Antibiotic controls were purchased from a variety of suppliers, predominantly Sigma-Aldrich (Castle Hill, Australia). Preparatory HPLC purification was conducted on an Agilent 1100 Prep using methanol:water mixtures. Analytical HPLC was conducted using a Phenomenex Luna C18 column (5 μm, 250 × 4.6 mm) and a flow of 1 mL min$^{-1}$ using a gradient of 100% 20:80 water: methanol to 100% methanol over 15 min, with UV detection at 230 nm. Cannabidiol (**1**) was supplied by Noramco. 7-hydroxy-1′,3′-diacetylcannabidiol (**A**), and 7-nor-7-carboxy-1′,3′-diacetylcannabidiol (**B**) were prepared following literature procedures[54,55].

### Triphenyl(3-phenylpropyl)phosphonium bromide.

A solution of triphenyl phosphine (12.5 g, 47.7 mmol) and 1-bromo-3-phenylpropane (50.2 mmol) in acetonitrile (250 mL) was heated to reflux for 16 h under an argon atmosphere. The reaction mixture was cooled to RT and the solvent displaced by toluene in vacuo to give a white solid 20.9 g (95%).

$^{1}$H NMR (CDCl$_3$): δ 1.97 (m, 2H), 3.05 (m, 2H), 3.94 (m, 2H), 7.21 (m, 5H), 7.68 (m, 7H), 7.78 (m, 8H).

### General procedure 1: Olivetol analog synthesis.

To a cold (−78 °C) suspension of the required phosphonium salt (25 mmol) in anhydrous THF (100 mL) was slowly added n-BuLi (2 M, 25 mmol) under an argon atmosphere. The cooling bath was removed, and the reaction was stirred at RT for 3 h then re-cooled to −78 °C. A solution of 3,5-dibenzyloxybenzaldehyde (25 mmol) in anhydrous THF (20 mL) was added dropwise and the temperature allowed to warm to RT overnight. The reaction was quenched by the addition of saturated ammonium chloride and the product extracted with ethyl acetate, dried (MgSO$_4$) and concentrated in vacuo. The crude material was purified by silica gel column chromatography (2% ethyl acetate:hexanes).

To a solution of the preceding compound (15 mmol) in ethanol (170 mL) was added, under an atmosphere of Argon, 10% Pd/C (0.25 equivalents). The Argon was displaced three times with Hydrogen and the reaction mixture was stirred overnight at RT. The solvent was removed in vacuo and the crude material was purified by silica gel column chromatography (30% ethyl acetate:hexanes).

*5-(4-Phenylbutyl)benzene-1,3-diol  (R = phenethyl).* Prepared following general procedure 1 with triphenyl(3-phenylpropyl)phosphonium bromide and 3,5-dibenzyloxybenzaldehyde, overall yield 79%.

$^{1}$H NMR (CDCl$_3$): δ 1.62 (m, 4H), 2.54 (m, 2H), 2.64 (m, 2H), 4.75 (br s, 2H), 6.19 (m, 1H), 6.24 (m,1H), 7.20 (m, 3H), 7.29 (m, 2H).

*5-Heptylbenzene-1,3-diol (R = pentyl).* Prepared following general procedure 1 with hexyl triphenylphosphonium bromide and 3,5-dibenzyloxybenzaldehyde, overall yield 81%.

$^{1}$H NMR (Solvent): δ 1.90 (t, 3H), 1.31 (m, 8H), 1.59 (m, 2H), 2.51 (m, 2H), 4.86 (br s, 2H), 6.19 (m, 1H), 6.26 (m,1H).

### General procedure 2: CBD and abn-CBD analog synthesis[56].

To a solution of the olivetol analog (5.1 mmol) in DCM (10 mL) was added zinc chloride (6.6 mmol) and water (25 mmol). The mixture was heated at reflux for 30 min whereby a DCM solution (5.0 mL) of (+)-p-mentha-2,8-dien-1-ol (5.1 mmol) was added dropwise over 30 min. Heating was continued for 2 h when TLC analysis indicated all (+)-p-mentha-2,8-dien-1-ol was consumed. The crude material was extracted by diethyl ether, washed (water, brine), dried (MgSO$_4$) and concentrated in vacuo. The desired products were purified by either silica gel column chromatography and/or preparative HPLC.

### Compounds MTC-007 (9) and MTC-018 (15) (R$_1$ = pentyl, R$_2$ = OH)

*MTC-007 (9).* Prepared following general procedure 2, with 5-heptylbenzene-1,3-diol (0.83 g, 4.0 mmol) and (+)-p-mentha-2,8-dien-1-ol (0.61 g, 4.0 mmol) and purified by column chromatography on silica gel using 5% ethyl acetate in hexane and further purified by preparative HPLC to give a yellow oil (0.20 g, 15%).

$^{1}$H NMR (CDCl$_3$): δ 0.88 (t, 3H), 1.29 (m, 8H), 1.55 (m, 2H), 1.66 (s, 3H), 1.79 (m, 5H), 2.09 (m, 1H), 2.23 (m, 1H), 2.42 (m, 3H), 3.86 (m, 1H), 4.56 (m, 1H), 4.66 (s, 1H), 4.78 (br s, 1H), 5.57 (s, 1H), 5.98 (br s, 1H), 6.19 (br s 1H), 6.26 (br s, 1H).

$^{13}$C NMR (CDCl$_3$): δ 14.20, 20.57, 22.77, 23,77, 28.54, 29.29, 29.37, 30.53, 31.07, 31.93, 35.65, 37.35, 46.30, 108.15, 109.83, 110.96, 113.92, 124.30, 140.10, 143.15, 149.40, 154.03, 156.20.

HRMS (M + H)$^{+}$: calculated for C$_{23}$H$_{35}$O$_2$:343.2637, found 343.2640.
HPLC: 99.0%.

*MTC-018 (15).* Prepared following general procedure 2, with 5-heptylbenzene-1,3-diol (0.83 g, 4.0 mmol) and (+)-p-mentha-2,8-dien-1-ol (0.61 g, 4.0 mmol) and purified by column chromatography on silica gel using 20% ethyl acetate in hexane to give an orange oil (0.38 g, 28%).

$^{1}$H NMR (CDCl$_3$): δ 0.88 (t, 3H), 1.30 (m, 4H), 1.46 (m, 2H), 1.53 (s, 3H), 1.79 (m, 5H), 2.09 (m, 1H), 2.24 (m, 1H), 2.48 (m, 1H), 2.59 (m, 1H), 3.53 (m, 1H), 4.46 (s, 1H), 4.64 (s, 1H), 5.03 (br m, 1H), 5.52 (s, 1H), 6.04 (s, 1H), 6.20 (s, 1H), 6.21 (s, 1H).

$^{13}$C NMR (CDCl$_3$): δ 14.20, 21.48, 22.79, 23.74, 28.27, 29.28, 29.79, 30.40, 31.53, 31.95, 34.16, 40.21, 45.11, 102.29, 108.74, 111.50, 120.00, 124.91, 139.85, 144.09, 147.83, 154.83, 156.59.

HRMS (M + H)$^{+}$: calculated for C$_{23}$H$_{35}$O$_2$:343.2637, found 343.2641.
HPLC: 98.9%.

### MTC-008 (16) and MTC-017 (10) (R$_1$ = phenethyl, R$_2$ = OH)

*MTC-008 (16).* Prepared following general procedure 2 with 5-(4-phenylbutyl) benene-1,3-diol (0.38 g, 1.6 mmol) and (+)-p-mentha-2,8-dien-1-ol (0.24 g, 1.6 mmol) and purified by column chromatography on silica gel using 33% ethyl acetate in hexane to give a pale yellow oil (0.25 g, 42%).

$^{1}$H NMR (CDCl$_3$): δ 1.50 (m, 5H), 1.64 (m, 3H), 1.77 (m, 5H), 2.07 (m, 1H), 2.19 (m, 1H), 2.28 (m, 1H), 2.45 (m, 1H), 2.62 (m, 3H), 3.47 (m, 1H), 4.43 (s, 1H), 4.62 (s, 1H), 4.73 (s, 1H), 5.44 (s, 1H), 6.01 (s, 1H), 6.15 (d, 1H), 6.19 (d, 1H), 7.16 (m, 3H), 7.25 (m, 2H).

$^{13}$C NMR (CDCl$_3$): δ 21.52, 23.75, 28.28, 30.40, 30.96, 31.43, 33.97, 35.88, 40.26, 45.07, 102.38, 108.73, 111.53, 120.09, 134.80, 125.81, 128.41, 128.55, 139.90, 142.58, 143.69, 147.86, 154.74, 156.67.

HRMS (M + H)$^{+}$: calculated for C$_{26}$H$_{33}$O$_2$:377.2481, found 377.2477.
HPLC: 99.6%.

*MTC-017 (10).* Prepared following general procedure 2 with 5-(4-phenylbutyl) benene-1,3-diol (1.0 g, 4.2 mmol) and (+)-p-mentha-2,8-dien-1-ol (0.63 g, 4.2 mmol) and purified by column chromatography on silica gel using 4% ethyl acetate in hexane to give a yellow oil (0.40 g, 25%).

$^{1}$H NMR (CDCl$_3$): δ 1.63 (m, 8H), 1.80 (m, 5H), 2.10 (m, 1H), 2.23 (m, 1H), 240 (td, 1H), 2.48 (t, 2H), 2.62 (t, 2H), 3.85 (m, 1H), 4.56 (s, 1H), 4.61 (br s, 1H), 4.66 (s, 1H), 5.57 (s, 1H), 5.97 (br s, 1H), 6.15 (br s, 1H), 6.25 (br s, 1H), 7.17 (m, 3H), 7.27 (m, 2H).

$^{13}$C NMR (CDCl$_3$): δ 20.62, 23.78, 28.55, 30.55, 31.10, 45.45, 35.92, 37.38, 46.29, 108.13, 109.93, 110.98, 114.02, 124.24, 125.75, 128.37, 128.55, 140.17, 142.74, 129.45, 154.05, 156.31.

HRMS (M + H)$^{+}$: calculated for C$_{26}$H$_{33}$O$_2$:377.2481, found 377.2479.
HPLC: 97.6%.

**MTC-011 (11) (R₁ = propyl, R₂ = OH)**. Prepared following general procedure 2 with 3-pentylphenol (WO2009019868) (0.61 g, 3.7 mmol) and (+)-*p*-mentha-2,8-dien-1-ol (0.57 g, 3.7 mmol) and purified by column chromatography on silica gel using 5% diethyl ether in hexane and further purified by preparative HPLC to give a pale yellow oil (0.22 g, 20%).

$^1$H NMR (CDCl$_3$): δ 0.89 (t, 3H), 1.31 (m, 4H), 1.57 (m, 2H), 1.59 (m, 5H), 1.77 (m, 5H), 2.07 (m, 1H), 2.21 (m, 1H), 2.52 (m, 2H), 3.41 (m, 1H), 4.57 (s, 1H), 4.67 (s, 1H), 5.42 (s, 1H), 5.52 (s, 1H), 6.63 (s, 1H), 6.87 (d, 1H).

$^{13}$C NMR (CDCl$_3$): δ 14.14, 20.95, 22.68, 23.74, 28.58, 30.53, 31.03, 31.69, 35.59, 43.75, 47.38, 111.03, 116.46, 120.54, 124.51, 127.15, 130.15, 137.32, 142.76, 148.72, 154.14.

HRMS (M + H)$^+$: calculated for C$_{21}$H$_{31}$O:299.2375, found 299.2377.
HPLC: 98.5%.

*MTC-009 (18)*. Prepared by hydrogenation[57] of compound **1**. Yellow oil, 75%

$^1$H NMR (CDCl$_3$): δ 0.87 (t, 6H), 0.90 (t, 3H), 1.33 (m, 4H), 1.40 (m, 1H), 1.60 (m, 4H), 1.79 (m, 4H), 2.13 (m, 2H0, 2.45 (m, 2H), 3.83 (, 1H), 4.88 (br s, 1H), 5.52 (s, 1H), 6.21 (br m, 3H).

$^{13}$C NMR (CDCl$_3$): δ 14.14, 16.53, 21.83, 22.27, 22.66, 23.72, 27.97, 30.79, 31.73, 35.64, 43.81, 107.63, 109.74, 114.16, 125.00, 140.11, 143.06, 154.56, 156.32.
HRMS (M + H)$^+$: calculated for C$_{21}$H$_{33}$O$_2$:317.2481, found 317.2491.
HPLC: 99.8%.

*MTC-014 (17)*. Prepared by hydrogenation[57] of compound **1**. Yellow oil, 91%

$^1$H NMR (CDCl$_3$): (major diastereoisomer) δ 0.71 (d, 3H), 0.85 (d, 3H), 0.90 (m, 6H), 1.07 (m, 2H), 1.31 (m, 4H), 1.61 (m, 7H), 1.80 (m, 1H), 2.03 (m, 1H), 2.42 (m, 2H), 3.02 (dt, 1H), 4.78 (s, 1H), 4.84 (s, 1H), 6.12 (s, 1H), 6.18 (s, 1H).

$^{13}$C NMR (CDCl$_3$): (major diastereoisomer) δ 14.14, 15.95, 21.82, 22.63, 22.67, 25.61, 28.78, 30.73, 31.75, 33.70, 35.44, 35.63, 38.33, 40.39, 44.80, 108.29, 109.21, 115.36, 142.04, 154.31, 155.68.
HRMS (M + H)$^+$: calculated for C$_{21}$H$_{35}$O$_2$:319.2637, found 319.2642.
HPLC: 99.8%.

*MTC-013 (13)*. Prepared by methylation[54,55] of compound **1**. Colorless oil, 89%.

$^1$H NMR (CDCl$_3$): δ 0.94 (t, 3H), 1.38 (m, 4H), 1.63 (m, 5H), 1.70 (s, 3H), 1.79 (m, 2H), 2.01 (m, 1H), 2.25 (m, 1H), 2.56 (m, 2H), 2.92 (m, 1H), 3.76 (s, 6H), 4.02 (m, 1H), 4.47 (m, 2H), 5.24 (s, 1H), 6.38 (s, 2H).

$^{13}$C NMR (CDCl$_3$): δ 14.07, 19.10, 22.57, 23.43, 29.75, 30.79, 30.99, 31.71, 36.17, 36.42, 45.25, 55.96, 105.07, 109.56, 119.07, 125.99, 131.14, 141.86, 149.53, 158.81.
HRMS (M + H)$^+$: calculated for C$_{23}$H$_{35}$O$_2$:343.2637, found 343.2637.
HPLC: 99.6%.

*MTC-012 (12)*. To a suspension of Cs$_2$CO$_3$ (1.6 g, 4.8 mmol) in anhydrous DMF (12 mL) was added compound **1** (1.0 g, 3.2 mmol) under an atmosphere of Argon at RT. Methyl iodide (0.20 mL, 3.2 mmol) was added dropwise and the reaction mixture was stirred for 2 h then quenched with water. The pH was adjusted to 8–9 and the product extracted with diethyl ether then washed with water and brine and dried (MgSO$_4$). Filtration and removal of the solvent in vacuo gave an orange syrup. The product was partially purified by silica gel column chromatography (1% ethyl acetate/hexane) then further purified by preparative HPLC to give a colorless oil (0.20 g, 19%).

$^1$H NMR (CDCl$_3$): δ 0.89 (t, 3H), 1.32 (m, 4H), 1.59 (m, 2H), 1.66 (s, 3H), 1.78 (m, 5H), 2.08 (m, 1H), 2.23 (m, 1H), 2.42 (m, 1H), 2.50 (m, 2H), 3.70 (s, 3H), 4.00 (m, 1H), 4.33 (s, 1H), 4.50 (s, 1H), 5.58 (s, 1H), 5.99 (br s, 1H), 6.22 (s, 1H), 6.31 (s, 1H).

$^{13}$C NMR (CDCl$_3$): δ 14.17, 18.88, 22.69, 23.81, 28.27, 30.51, 30.95, 31.70, 35.64, 36.16, 46.80, 55.69, 103.29, 109.68, 111.01, 115.21, 124.70, 139.64, 142.81, 147.42, 155.85, 158.31.
HRMS (M + H)$^+$: calculated for C$_{22}$H$_{33}$O$_2$:329.2481, found 329.2480.
HPLC: 99.0%.

*MTC-002 (5)*.

B

To a solution of compound **3** (0.69 g, 1.6 mmol) in anhydrous DMF (46 mL) under an atmosphere of Argon, was added *N,N*-diisopropylethylamine (0.67 mL, 3.9 mmol), HATU (0.92 g, 2.4 mmol) and methylamine hydrochloride (0.22 g, 3.2 mmol). The reaction mixture was stirred for 1 h 40 min at RT and then poured into water and extracted twice with ethyl acetate. The combined organic layers were washed with water and brine, dried (MgSO$_4$), filtered and the solvent was

removed in vacuo. The residue was purified by silica gel column chromatography (60% ethyl acetate:hexane) to give a yellow gum (0.68 g, 96%).

$^1$H NMR (CDCl$_3$): δ 0.88 (t, 3H), 1.33 (m, 4H), 1.62 (m, 5H), 1.69 (m, 1H), 1.92 (m, 1H), 2.24 (br s 7H), 2.58 (m, 2H), 2.76 (m, 2H), 2.83 (d, 2H), 3.60 (m, 1H), 4.53 (s, 1H), 4.63 (s, 1H), 5.96 (br s, 1H), 6.01 (t, 1H), 6.76 (br s, 2H).

The preceding compound (0.82 g, 1.9 mmol) was dissolved in ethanol (42 mL) and sodium borohydride (93 mg, 2.5 mmol) was added. The mixture was heated to reflux for 3 h after which time further sodium borohydride (80 mg, 2.1 mmol) was added and heating was continued for 1 h. The reaction mixture was cooled to RT and the solvent removed in vacuo and the residue was partitioned with ethyl acetate and water. The pH was adjusted to 2 with 2 M HCl and the organic layer was washed with brine, dried (MgSO$_4$), filtered and the solvent was removed in vacuo. The residue was partially purified by silica gel column chromatography (70% ethyl acetate:hexane) and then further purified by preparative HPLC to give a white solid (130 mg, 19%)

$^1$H NMR (CD$_3$OD): δ 0.89 (t, 3H), 1.32 (m, 4H), 1.55 (m, 2H), 1.65 (s, 3H), 1.72 (m, 2H), 1.87 (m, 1H), 2.38 (m, 4H), 3.07 (m, 1H), 4.05 (m, 1H), 4.48 (s, 1H), 6.10 (s, 2H). 6.47 (s, 1H).

$^{13}$C NMR (CD$_3$OD): δ 14.37, 19.46, 23.57, 260.03, 26.52, 30.45, 31.98, 32.58, 36.60, 38.01, 45.46, 108.10, 110.83, 114.60, 131.50, 140.85, 143.21, 149.87, 157.63, 172.36.
HRMS (M + Na)$^+$: calculated for C$_{22}$H$_{31}$NNaO$_3$:380.2202, found 380.2195.
HPLC: 100.0%.

*MTC-005 (3)*.

A

To a solution of compound **2** (0.53 g, 1.3 mmol) in anhydrous DCM (10 mL) under an atmosphere of Argon was added boron trifluoride etherate (5.0 μL, 0.038 mmol). The reaction mixture was cooled to 0 °C and trimethylsilyldiazomethane (2.0 M in hexane, 0.64 mL, 1.3 mmol) was added dropwise. After 30 min, additional boron trifluoride etherate (3.0 μL, 0.024 mmol) and trimethylsilyldiazomethane (0.15 mL, 0.30 mmol) were added and the reaction was allowed to warm to RT over 45 min. The reaction was quenched with water and extracted with dichloromethane and the organic layer was dried (MgSO$_4$), filtered and the solvent removed in vacuo. The residue was purified by silica gel column chromatography (6% ethyl acetate:hexane) to give a gum (0.14 g, 26%).

$^1$H NMR (CDCl$_3$): δ 0.88 (t, 3H), 1.31 (m, 4H), 1.58 (m, 5H), 1.73 (m, 1H), 1.85 (m, 1H), 2.21 (br s 8H), 2.55 (m, 1H), 2.66 (t, 1H), 3.30 (s, 3H), 3.56 (m, 1H), 3.74 (d, 1H), 3.80 (d, 1H), 4.44 (s, 1H), 4.57 (s, 1H), 5.49 (s, 1H), 6.76 (s, 2H).

To a solution of the preceding compound (0.15 g, 0.35 mmol) in ethanol (20 mL) was added sodium borohydride (0.018 g, 0.47 mmol) under an atmosphere of Argon. The reaction was heated to reflux for 1.25 h and then cooled to RT and the solvent removed in vacuo. The residue was partitioned with ethyl acetate and water and the pH was adjusted to 4 with 1 M HCl and the organic layer was washed with brine, dried (MgSO$_4$), filtered and the solvent was removed in vacuo. The residue was partially purified by silica gel column chromatography (12% ethyl acetate: hexane) and then further purified by preparative HPLC to give a colorless oil (0.080 g, 66%).

$^1$H NMR (CDCl$_3$): δ 0.88 (t, 3H), 1.30 (m, 4H), 1.56(m, 2H), 1.67 (s, 3H), 1.79 (m, 1H), 1.88 (m, 1H), 2.44 (m, 2H), 2.49 (m, 1H), 3.32 (s, 3H), 3.88 (m, 2H), 3.94 (m, 1H), 4.56 (s, 1H), 4.65 (s, 1H), 4.72 (br s, 1H), 5.59 (br s, 1H), 5.84 (s, 1H), 6.21 (br s, 2H).

$^{13}$C NMR (CDCl$_3$): δ 14.15, 20.35, 22.67, 26.36, 28.41, 30.74, 31.64, 35.61, 36.99, 46.54, 58.15, 76.32, 108.29, 109.90, 111.13, 113.53, 127.42, 139.82, 143.30, 149.05, 154.08, 155.93.
HRMS (M + Na)$^+$: calculated for C$_{22}$H$_{33}$NaO$_3$: 367.2249, found 367.2253.
HPLC: 97.2%.

*MTC-015 (cannabidivarin, 6)*[2]. $^1$H NMR (CD$_3$OD): δ 089 (t, 3H), 1.56 (m, 2H), 1.64 (s, 3H), 1.68 (s, 3H), 1.74 (m, 2H), 2.00 (m, 1H), 2.20 (m, 1H), 2.36 (m, 2H), 3.93 (m, 1H), 4.45 (m, 2H), 4.80 (s, 1H), 5.29 (s, 2H).

$^{13}$C NMR (CD$_3$OD): δ 12.77, 18.16, 22.337, 23.96, 29.34, 30.33, 36.12, 37.43, 45.04, 107.05, 109.12, 114.64, 125.88, 133.04, 141.09, 148.91, 156.05.
HRMS (M + H)$^+$: calculated for C$_{19}$H$_{27}$O$_2$: 287.2013, found 287.2011.
HPLC: 98.7%
*MTC-016 (7-nor-7-carboxy-cannabidivarin, 8)* was prepared adapting reports[1] in the literature.

$^1$H NMR (CD$_3$OD): δ 0.91 (t, 3H), 1.57 (m, 2H), 1.65 (s, 3H), 1.72 (m, 1H), 1.83 (m, 1H), 2.25 (m, 1H), 2.37 (t, 2H), 2.45 (m, 1H), 3.03 (m, 1H), 4.05 (m, 1H), 4.48 (m, 2H), 6.09 (s, 2H), 6.86 (s, 1H).

$^{13}$C NMR (CD$_3$OD): δ 12.73, 18.03, 23.93, 24.58, 29.03, 37.00, 37.46, 44.07, 106.68, 109.47, 112.97, 127.33, 141.65, 146.30, 148.44, 156.21, 170.22.

HRMS (M + Na)$^+$: calculated for C$_{19}$H$_{24}$NaO$_4$: 339.1572, found 339.1573.

HPLC: 100%

*MTC-019 (7-nor-7-hydroxymethyl-cannabidivarin,* **7***)* was prepared adapting reports[1] in the literature.

$^1$H NMR (CD$_3$OD): δ 0.91 (t, 3H), 1.58 (m, 2H), 1.66 (s, 3H), 1.77 (m, 1H), 1.82 (m, 1H), 2.21 (m, 2H), 2.38 (t, 2H), 3.05 (m, 1H), 3.96 (m, 2H), 4.01 (m, 1H), 4.49 (m, 2H), 5.55 (s, 1H), 6.12 (s, 2H).

$^{13}$C NMR (CD$_3$OD): δ 12.73, 18.12, 23.94, 25.89, 29.28, 35.98, 37.41, 44.89, 66.19, 107.03, 109.07, 114.40, 127.68, 135.67, 141.17, 148.96, 156.09.

HRMS (M + Na)$^+$: calculated for C$_{19}$H$_{26}$NaO$_3$: 325.1780, found 325.1766.

HPLC: 98.3%

**Determination of antimicrobial activity**. Antimicrobial activity of CBD was tested against a number of bacterial strains by broth microdilution (BMD) minimum inhibitory concentration (MIC) assays using procedures described by Clinical and Laboratory Standards Institute (CLSI)[58–61]. MICs were conducted at the University of Queensland (UQ), MicroMyx LLC, and Monash University. ATCC strains were sourced from the American Type Culture Collection and NRS strains from NARSA (Network on Antimicrobial Resistance in *Staphylococcus aureus*) via BEI Resources (www.beiresources.org), with clinical MSSA and MRSA isolates from Prof Graeme Nimmo, Queensland Health Central Pathology. CBD was prepared as a 10 mg mL$^{-1}$ stock solution in DMSO, then diluted to 2.56 mg mL$^{-1}$.

### MIC assays

*UQ procedure. Aerobic assays*: The compounds, along with standard antibiotics were serially diluted two-fold in cation-adjusted Mueller Hinton broth (CAMHB) (Bacto laboratories, Cat. No. 212322) for bacteria assay and Yeast Nitrogen Base (YNB) (BD, Cat. No. 233520) for fungi (yeast) assays across the wells of flat bottom 96-well polystyrene plates (Corning, Cat. No. 3370). Standard antibiotic comparators ranged from 0.03–64 µg mL$^{-1}$ for bacteria, and 0.06–128 µg mL$^{-1}$ for fungi, and from 0.03–64 µg mL$^{-1}$ for CBD. Bacteria were cultured in CAMHB at 37 °C overnight. A sample of each culture was then diluted 40-fold in fresh CAMHB and incubated at 37 °C for 2–3 h. The resultant mid-log phase cultures were added to the compound-containing 96-well plates to give a final cell density of 5 × 10$^5$ CFU mL$^{-1}$. Fungal strains were cultured on Yeast extract Peptone Dextrose (YPD) agar at 30 °C for 72 h. Single colonies were taken from the agar plate and dissolved in sterile water, then the solution was adjusted in YNB media giving a final cell density of 5 × 10$^3$ CFU mL$^{-1}$.

All the plates were covered and incubated at 37 °C for 20 h for bacteria and at 35 °C for 36 h for fungi. MICs were the lowest concentration showing no visible growth. For comparisons of different plate types, MICs were conducted in polystyrene (Corning, Cat. No. 3370), polypropylene (Corning, Cat. No. 3364) and non-binding surface (NBS) polystyrene (Corning, Cat. No. 3641) 96-well plates and it was deemed that polystyrene was the most suitable for the CBD compound class.

*Anaerobic Assays*: All steps were performed in a COY type B anaerobic chamber with the anaerobic atmosphere controlled by the introduction of 10%CO$_2$/5% H$_2$ in N$_2$CoA gas mix, catalyst Stak-Pak and O$_2$-H$_2$ gas analyzer, with H$_2$ levels kept at ~2% for the duration of the assay. Brain Heart Infusion (BHI) broth (Oxoid CM1135B) media with 1% cysteine to further promote an anaerobic environment was used for this assay and incubated in the anaerobic chamber for 24 h prior to use for reduction of oxygen.

CBD and control antibiotics were serially diluted in BHI, two-fold across the wells of 96-well of polystyrene 96-well plates (Corning, Cat. No. 3370). Plates were set up in duplicate for each strain. All bacteria strains were cultured on Tryptic Soy agar (TSA, BD, Cat. No. 236950) at 37 °C for 72 h. Several colonies were taken from the agar plate and dissolved in BHI broth. The solution was then adjusted to OD$_{600}$ 0.5–0.7 and diluted to a final cell density of 5 × 10$^5$ CFU mL$^{-1}$, 100 µL were added to the test plate, giving a final CBD concentration range of 0.06–128 µg mL$^{-1}$. All the plates were covered and incubated at 37 °C for 48 h. The MIC was defined as the lowest concentration with which no growth was visible after incubation.

*M. tuberculosis H37Rv*. The MIC study was performed using a resazurin reduction microplate assay as previously described[62]. For normoxic conditions, the compound containing plates (prepared in the same manner as previously described for aerobic bacteria BMD MIC assay) were incubated for 5 days at 37 °C in a humidified incubator prior to the addition of 30 µL of a 0.02% resazurin solution and 12.5 µL of 20% Tween-80 to each well. After 24 h incubation (37 °C), sample fluorescence was measured on a Fluorostar Omega fluorescent plate reader (BMG) with an excitation wavelength of 530 nm and emission read at 590 nm. Percent fluorescence relative to the positive control wells (H37Rv without compound) minus the negative control wells (media only) was plotted for the determination of the MIC (≤90% reduction in growth). The assays were performed in replicate on independent occasions (*n* = 3–6).

*Micromyx procedure*. MIC values were determined using a broth microdilution procedure described by Clinical and Laboratory Standards Institute (CLSI)[58–61].

*Test media*: Prior to testing, aerobic bacteria and *Campylobacter* were streaked from frozen vials onto Tryptic Soy Agar (TSA) with 5% sheep blood (BD, Lot No. 9192895), anaerobic bacteria were streaked onto Supplemented Brucella Agar (SBA; BD, Lot No. 9219195), and *Legionella* was grown on CYE Agar (Oxoid, Lot No. 2280778) supplemented with *Legionella* Growth Supplement BCYE (HiMedia, Lot No. 0000337458). *Haemophilus* and *Neisseria* were streaked onto Chocolate agar (BD, Lot No. 9228071). Aerobic bacteria were incubated at 35 °C overnight, *Campylobacter* was incubated in microaerophilic conditions for 48 h, *Streptococcus*, *Haemophilus*, and *Neisseria* were incubated at 35 °C in 5% CO$_2$ for 24 h, and anaerobic bacteria were incubated anaerobically at 35 °C for 48 h.

Cation-adjusted Mueller-Hinton broth (CAMHB; BD, Lot No. 8190586) was used for MIC testing of aerobic organisms. For strains of *Streptococcus*, *L. monocytogenes*, *Corynebacterium spp.*, and *C. jejuni*, this medium was supplemented with 3% lysed horse blood (LHB; Hemostat, Dixon, CA; Lot No. 474990). For testing of *Haemophilus*, Haemophilus Test medium (HTM) was prepared by supplementing CAMHB with 15 µg mL$^{-1}$ NAD (Sigma Aldrich, Lot No. SLBX4629), 15 µg mL$^{-1}$ hematin (Sigma Aldrich, Lot No. SLBD4979V) and 5 g L$^{-1}$ yeast extract (Sigma Aldrich, Lot No. SLBD4979V). Brucella broth (BD; Lot No. 7128995) supplemented with hemin (Sigma; Lot No. SLBC4685V), Vitamin K1 (Sigma; Lot No. MKCG2073), and 5% LHB was used for MIC testing of anaerobic organisms.

For *Neisseria* broth assays a modified medium described by the ATCC as capable of supporting growth was used. This medium contains 15 g Oxoid Special Peptone, 1 g corn starch (Ward's Science; Rochester, NY; Lot 39–3271), 5 g NaCl (VWR; 57897), 4 g K$_2$HPO$_4$ (Sigma Aldrich, Lot No. SLBT7061), 1 g KH$_2$PO$_4$ (SLBC1921V), and 1% IsoVitaleX (BD, Lot No. 8323954) enrichment per 500 mL.

Buffered Yeast Extract Broth (BYEB) was prepared to test the Legionella isolates. For 1 L of this medium, 10 g H$_2$NCOCH$_2$NHCH$_2$CH$_2$SO$_3$H (ACES; Alfa Aesar, Haverhill, MA; Lot No. 10185658), 10 g yeast extract (Oxoid, Lot No. 1424987–02), 1 g alpha ketoglutarate (Sigma Aldrich, Lot No. BCBX4691), 0.4 g L-cysteine (Sigma Aldrich, Lot. No. BCBQ4116V) and 0.25 g Iron (III) pyrophosphate (Sigma Aldrich, Lot No. SLBK9182V) were dissolved in H$_2$O, pH-adjusted to 6.9 using 1 N NaOH, and filter-sterilized using a 0.22 µm filter.

*Assay procedure*. In mother plates, the wells of columns 2–12 of standard 96-well microdilution plates (Costar 3795) were filled with 150 µL of the designated diluent for each row of drug. The test article and comparator compounds (300 µL at 101× the highest concentration to be tested) were dispensed into the appropriate wells in column 1. Two-fold serial dilutions were then made in the mother plates from columns 1–11. The wells of column 12 contained no drug and served as the organism growth control wells for the assay.

Daughter plates were loaded with 190 µL per well of the appropriate test medium for the tested organism. Two microliter of drug solution from each well of a mother plate was transferred to the corresponding well of each daughter plate. Daughter plates for the testing of anaerobes pre-reduced in a Bactron II anaerobe chamber for 2 h prior to inoculation. A standardized inoculum of each test organism was prepared per CLSI methods[58–61]. Plates were inoculated with 10 µL of the inoculum resulting in a final cell density of approximately 5 × 10$^5$ CFU mL$^{-1}$ per well. Additional information on the test media and used for each bacteria and their preparation is included in Supplementary material, including details for the agar dilution assays that were also used for *N. gonorrhoeae*.

Plates were incubated at 35 °C for approximately 16–20 h (aerobes), 20–24 h (*Streptococcus*, *Neisseria*, *Haemophilus*, *Listeria*, and *Corynebacterium*), 48 h (*Campylobacter*), and both 24 and 48 h for *Legionella* as specified by CLSI[58–61]. Anaerobe plates were placed in a BD GasPak EZ Anaerobe Container System and incubated at 35 °C for 48 h. Following incubation, microplates were removed from the incubator and viewed from the bottom using a plate viewer. The MIC was read and recorded as the lowest concentration of drug that inhibited visible growth of the organism. All assays were done with triplicate independent inocula, except for the *N. gonorrhoeae* broth assays (*n* = 1).

*C. difficile procedure (Monash)*. Five strains of *C. difficile*, implicated in both human and animal infections, were grown and re-streaked out to single colonies on Heart infusion (HIS) agar plates supplemented with sodium taurocholate, glucose and L-cysteine (3.7% heart infusion broth (Oxoid), 0.5% yeast extract (Oxoid), 1.5% agar (Difco), 0.1% sodium taurocholate (New Zealand Pharmaceuticals), 0.375% glucose and 0.1% L-cysteine HCl (Sigma-Aldrich)). 3–5 well isolated colonies were used to inoculate fresh, pre-reduced supplemented HIS broth (3.7% heart infusion broth (Oxoid), 0.5% yeast extract (Oxoid), 0.375% glucose and 0.1% L-cysteine HCl (Sigma-Aldrich)) and grown overnight anaerobically (10% H$_2$, 10% CO$_2$, 80% N$_2$) at 37 °C. Overnight cultures were inoculated into fresh supplemented HIS and grown again anaerobically at 37 °C to an OD$_{600}$ of ~1.4–1.5. Two 10-fold dilutions of each culture were performed in pre-reduced fresh, sterile supplemented HIS to obtain an approximate desired inoculum concentration. CBD was dissolved in DMSO to a concentration of 10 mg mL$^{-1}$. This was used to spike fresh supplemented HIS broth to a final concentration of 256 µg mL$^{-1}$ (double the highest concentration we subjected *C. difficile* to, as addition of an equal volume of bacterial inoculum to the MIC plate dilutes the CBD concentration by half). A 2-fold serial dilution of the 256 µg mL$^{-1}$ CBD stock was performed out to 2 µg mL$^{-1}$ (final minimum CBD concentration tested on *C. difficile* hence 1 µg mL$^{-1}$). A

96-well plate was prepared with strains tested in duplicate and 100 μL of the CBD concentrations loaded into wells 1 through 8 from highest concentration to lowest. Sterility and growth controls (100 μL of fresh Supplemented HIS) in wells 12 and 10, respectively, as well as DMSO controls (100 μL 2.5% DMSO in Supplemented HIS) were also prepared. One hundred microliters of each respective *C. difficile* strain inoculum was added to each well (except those denoting sterility controls) of the pre-reduced, CBD inoculated 96-well plate and the entire plate left to incubate anaerobically at 37 °C for 24 h. The protocol was repeated 3 times to produce 3 biological replicates each with 2 technical replicates. MICs were determined visually as the lowest concentration at which there was no bacterial growth. This was further confirmed by spot plating certain wells of the MIC assays as detailed below.

Addition of CBD dissolved in DMSO to Supplemented HIS caused a turbid/cloudy appearance at higher concentrations. It was important to discern whether this was distinct from the turbidity observed due to bacterial growth in growth controls and non-inhibited CBD concentrations post incubation. To do this 50 μL samples of the 128 μg mL$^{-1}$, 64 μg mL$^{-1}$, MIC concentration, one below the MIC concentration, growth control and sterility control were taken for each of the five *C. difficile* strains following plate incubation *via* vigorous pipetting to dislodge and homogenize any bacteria present. Samples were heat shocked at 65 °C for 30 min to stimulate *C. difficile* sporulation then 20 μL spot plated onto HIS-taurocholate agar and grown overnight at 37 °C under anaerobic conditions. The presence or absence of *C. difficile* colonies was assessed after 24 h.

**Neisseria gonoreheae agar dilution MIC assay (Micromyx).** MIC values were determined using the agar dilution method. All serial dilutions and liquid handling were performed by hand using sterile pipettes. Ciprofloxacin and ceftriaxone were prepared as stock solutions at 40× in the appropriate solvent and diluted according to CLSI guidelines. Cannabidiol was prepared as a stock solution at 100× the final concentration in the appropriate solvent and diluted according to CLSI guidelines.

Each test agent was mixed with molten (50–55 °C) GC media agar supplemented with 1% IsoVitalex. Ciprofloxacin and ceftriaxone were added to the medium in a ratio of 2 mL 10× test agent to 18 mL agar. All other test agents were added to the medium in a ratio of 0.2 mL 100× test agent to 19.8 mL agar. Once test agents were added to the agar in a sterile tube, they were mixed gently and then poured into a sterile 110 × 15 mm petri dish. Plates were allowed to solidify at room temperature and placed in a laminar air flow hood with the covers off to remove condensed moisture on the agar surface.

Next, each isolate was suspended to the equivalent of a 0.5 McFarland standard in saline using a Siemens Microscan turbidity meter and diluted 1:10. Each bacterial cell suspension was then transferred to wells in a stainless-steel replicator block. The prongs on the replicator deliver approximately 1–2 μL of inoculum to an agar surface. The resulting inoculum spots contained approximately 10$^4$ cells/spot.

Each agar plate containing either test compound or no drug (control) was stamped with the replicator. All plates were placed with the agar surface up to allow for the inoculum to soak into the agar. The plates were then inverted and incubated at 35 °C for 24 h in a CO$_2$ incubator and finally inspected for growth. The MIC was defined as the lowest test agent concentration that substantially inhibited bacterial growth on the agar surface.

**Serum reversal MIC assay.** The CBD compounds and standard antibiotics of concentration range 0.03–64 μg mL$^{-1}$ were tested in a BMD MIC assay, as described in the UQ MIC assay section above, in the presence of a mixture of 50% human serum (Sigma-Aldrich, Cat. No. H3667-100ML) and 50% CAMHB (BD, Cat. No. 212322).

**Agar disc diffusion assay.** Final CBD concentrations between 5–175 μg were added to sterile blank paper discs (6 mm) (*n* = 2 discs per concentration), and the discs were then allowed to dry in a biosafety cabinet. *S. aureus* ATCC 43300 (MRSA) was cultured in Mueller Hinton broth (MHB) (Bacto laboratories, Cat. No. 211443) at 37 °C overnight. A sample of each culture was then diluted 40-fold in fresh MHB and incubated at 37 °C for 2–3 h. The inoculum was then diluted 40-fold in saline water to give a mid-log phase culture, and 100 μL was spread evenly onto the surface of a Mueller Hinton agar plate. The CBD-loaded disc was then placed on the plate and 10 μL of distilled water added to each disc. A blank disc was placed at the center of the plate for control. The zone of inhibition was the diameter of the clear zone around the loaded discs and was measured in mm. No observation of clear zone was recorded as 6 mm in diameter, which was the same as the diameter of paper disc.

**BMD MIC synergy assays.** Cannabidiol was tested in combination with polymyxin B and colistin in a 9 × 16 matrix in 384-well polystyrene plates (Corning; Cat. No. 3680). Each plate contained the antibiotics in a 16 point 2-fold serial dilution down the wells of the 384 well plate (final test concentration ranging from 16–0.0019 μg mL$^{-1}$) followed by a 9 point 2-fold serial dilution of CDB across the wells (final test concentration ranging from 64–0.25 μg mL$^{-1}$). Each well containing 1:1 combination of concentration of both antibiotic and CBD. Each plate contained the antibiotics and CBD plated individually as 2-fold serial dilution to determine the baseline MIC values of each compound when tested alone.

Bacteria were cultured in CAMHB (BD, Cat. No. 212322) at 37 °C overnight, then diluted 40-fold and incubated at 37 °C for a further 2–3 h. The resultant mid-log phase cultures were diluted in CAMHB and added to each well of the compound-containing 384-well plates to give a final cell density of $5 \times 10^5$ CFU mL$^{-1}$ in 50 μL test solution per well. The plates were incubated at 37 °C for 18–20 h. Optical density was read at 600 nm (OD$_{600}$) using Gen5 Spectrophotometer. MIC was determined as the lowest concentration at which OD$_{600}$ demonstrated ≥90% growth inhibition compared to growth control. Analysis was performed using Microsoft Excel.

The synergistic effect was calculated based on the equation:

$$\text{FICI} = \frac{\text{MIC compound A in combination}}{\text{MIC compound A alone}} + \frac{\text{MIC compound B in combination}}{\text{MIC compound B alone}}$$

Where synergy is defined as a fractional inhibitory concentration index (FICI) ≤ 0.5.

**Time-kill assay.** BMD MIC plate was prepared as previously described (MIC assay, UQ procedure) giving a final cell density $5 \times 10^5$ CFU mL$^{-1}$ *S. aureus* ATCC 43300 (MRSA) and compound concentration range of 0.03–64 μg mL$^{-1}$ incubated at 37 °C. Serial dilutions of the cultures were plated at each desired time point, using a multichannel pipette as follows: in row A of a 96-well plate, 50 μL of sterile activated charcoal suspension (25 mg mL$^{-1}$) was added, while 90 μL of 0.9% sterile saline per well was added to the other rows. At selected time points, 50 μL aliquots were transferred from the time-kill assay plate to the first row containing the charcoal suspension and mixed well. The wells were further diluted 1 in 10 (0.9% saline) for the appropriate number of dilutions and 10 μL of each dilution was spotted in duplicate onto Luria-Bertani (LB) agar, then incubated overnight at 37 °C. The colonies in each spot were counted and used to calculate the number of viable CFU mL$^{-1}$ remaining in the original culture by considering the dilution factors (1:2 in charcoal, the serial dilution factor and the volume of the aliquot spotted).

**Biofilm assay.** *S. aureus* ATCC 25923 (MSSA) and *S. aureus* ATCC 43300 (MRSA) were cultured in Tryptic Soy Broth (TSB) (BD, Cat. No. 211825) at 37 °C overnight. Bacteria were then diluted 1:100 in TSB with 3% glucose and 100 μL added to the wells of 96-well polystyrene plates (Corning; Cat. No. 3370), leaving row A and H as media only controls. Plates were plated in duplicate and incubated at 37 °C for 48 h to allow for biofilm formation.

Standard antibiotic comparators and CBD were tested as a 12-point dose response from 0.03–64 μg mL$^{-1}$. Standard antibiotics and CBD were serially diluted in TSB with 3% glucose two-fold across the wells of 96-well polystyrene plates (Corning; Cat. No. 3370), to a concentration range of 0.06–128 μg mL$^{-1}$, with 100 μL final volume (antibiotic plate). Following the 48 h incubation, the biofilm-containing plates were carefully washed three times with 200 μL per well of saline solution (0.9% NaCl, Baxter Healthcare; Cat. No. AHF7124) using a manual pipette to remove the planktonic cells but leaving the biofilm uninterrupted. Then, from the antibiotic plate 100 μL was transferred into the washed biofilm containing plates and incubated at 37 °C for a further 24 h. After incubation, plates were washed three times with saline solution, then fixed with 100 μL per well of 99% methanol for 15 min. Once the biofilm was fixed, 100 μL per well 0.1% crystal violet biological stain (Sigma; Cat no. C0775–25G) was added for 20 min and used as an indicator for biofilm formation, followed by three saline solution washing and air drying cycles. To dissolve the crystal violet, 150 μL per well of methanol was added to allow for minimum biofilm eradication concentration (MBEC) analysis. The biofilm formation was determined by absorbance read at OD$_{590}$ on Tecan M1000 Pro spectrophotometer. MBIC was determined as the lowest concentration at which OD$_{590}$ demonstrated ≥70% growth inhibition compared to growth control. Analysis was performed using Microsoft Excel.

**Biofilm microscopy assay.** CBD concentrations ranged from 1–128 μg mL$^{-1}$. *S. aureus* ATCC 43300 (MRSA) was cultured in TSB (BD, Cat. No. 211825) at 37 °C overnight. Samples were then diluted 1:100 in fresh TSB supplemented with 3% glucose. Two hundred microliter of the bacteria suspension was added onto coverslips across a 96-well polystyrene plate (Corning, Cat. No. 3370), incubated at 37 °C for 48 h to generate the biofilm. CBD was diluted in TSB with 3% glucose. After incubation, media was removed from the biofilm-containing plates and the plates were carefully washed once with 200 μL well$^{-1}$ of saline solution (0.9% NaCl, Baxter Healthcare; Cat. No. AHF7124). Then, 200 μL of CBD was transferred into the washed plates containing the biofilm and incubated at 37 °C for 24 h. After incubation, supernatant was removed, and plates were washed once with saline solution. A TSB solution (150 μL) containing PI (10 μM) and SYTO 9 (10 μM) was then dropped onto the coverslips, left for 1 h at room temperature, then washed once with saline solution followed by mounting on slides using VectaShield H1000 as a mounting media. Coverslips were investigated by confocal microscopy using a Confocal 1-Leica SP8 STED (HC PL APO CS2 40×/1.10 WATER). Images were processed and counted with Fiji (Image J).

**Resistance frequency assay.** The resistance frequency for CBD was determined following literature procedures[63,64]. In this assay, final CBD concentrations ranged

from 1–16 µg mL$^{-1}$. S. aureus ATCC 43300 (MRSA) was cultured in CAMHB (BD, Cat. No. 212322) at 37 °C overnight. Bacteria were then serially diluted to a $10^{-8}$ dilution. To determine the titer of the culture, the dilutions $10^{-6}$, $10^{-7}$, and $10^{-8}$ were used. One hundred microliter of these dilutions were added to compound-free petri plates containing TSB (BD, Cat No. 211825) with 1% agarose, plated in duplicate. Plates were incubated overnight at 37 °C. CBD was added to 160 mL TSB with 1% agarose in a 200 mL glass bottle maintained at 45–50 °C in a water bath, giving a final concentration 16× above MIC. Then, 80 mL of agar was plated in 4 petri plates (duplicates/condition – Neat, $10^{-1}$) and swirled to coat the bottom of the plate equally. The remaining compound containing agar was then serially diluted two-fold by adding an extra 80 mL of agarose. Again, 80 mL of agar were plated in 4 petri plates. This procedure was repeated until 1 × or 2 × MIC value was reached. One hundred microliter of neat culture and $10^{-1}$ dilution (stationary phase) were plated onto agarose containing CBD at 16×, 8×, 4×, 2×, and 1× MIC values. Plates were incubated overnight at 37 °C for 24 h. After incubation, colony counts were completed. The well-separated visible colonies were counted; any colonies that grew together were not included in the count. Resistance frequency was determined as follows:

$$\text{Resistance frequency} = \frac{\frac{[(\text{Average neat values}) + (\text{Average } 10^{-1} \text{ values}) \times 10]}{4(\text{duplicate no.})}}{\text{Average titer of culture per mL}(2.360 \times 10^{11}) \times \text{Amount plated}(0.1 \text{ mL})}$$

**Broth dilution serial passage resistance induction assay.** A resistance induction (20-day generational passage) assay was conducted for CBD and daptomycin[65]. In this assay, final CBD concentrations ranged from 0.25–5 µg mL$^{-1}$ and final daptomycin concentrations ranged from 0.094–2 µg mL$^{-1}$. A mid-log S. aureus (MRSA, ATCC 43300) growth culture was serially diluted and plated on TSA plates in duplicate and incubated at 37 °C overnight to determine viable colony count.

*Day 1.* CBD and daptomycin were diluted in CAMHB (BD, Cat. No. 212322) and 100 µL of the dilutions added to columns 1–10 across rows of 96-well polystyrene plates (Corning, Cat. No. 3370). Column 11 was used as the drug-free passage well, and column 12 as a negative growth control with 200 µL uninoculated media in each well. After overnight culture, MRSA samples were diluted 40-fold and incubated at 37 °C for a further 2–3 h. The resultant mid-log phase cultures were diluted in CAMHB and 100 µL added to each well of the compound-containing 96-well plates to give a final cell density of $5 \times 10^5$ CFU mL$^{-1}$. The plate was covered and incubated at 37 °C for 20 h.

*Days 2–20.* MICs were determined visually after overnight incubation, defined as the lowest concentration at which no growth was visible after incubation. CBD and daptomycin tested concentrations were established to ensure at least three concentrations above, and three concentrations below MIC, based on the previous MIC results. Compound-containing media was added, 100 µL to each well of plate.

For each replicate, the well with the highest concentration of drug permitting growth was selected by visual inspection. The bacterial growth in those wells was resuspended by pipetting, then plates were read for OD$_{600}$, which was used to calculate the dilution of cells to a density of $10^6$ CFU mL$^{-1}$. Bacteria were diluted in CAMHB and 100 µL added to each well of the next replicate passage. The final well volume was 200 µL with a cell density of $5 \times 10^5$ CFU mL$^{-1}$.

Once prepared, the plate was covered and incubated at 37 °C overnight. Plate reading, compound preparation and bacterial preparation were repeated from Day 2 to Day 20.

*Drug free passages.* Following 20 days of passaging in the presence of CBD and daptomycin, each replicate was passaged for four days in drug-free media to assess the stability of any induced resistance. The Day 20 plate was read and the same bacterial preparation methodology was followed to prepare the additional four days of passaged bacterial samples free of CBD and daptomycin.

**Membrane depolarization assay (HD Biosciences).** Antibiotic-induced bacterial cytoplasmic membrane depolarization was determined using the fluorescent dye 3,3-dipropylthiacarbocyanine DiSC3(5) (Sigma Aldrich, Cat. No. 43608) in a similar manner to that previously described[66]. Bacteria were prepared by streaking from single-use frozen vials of S. aureus ATCC 29213 or E. coli SPT-39 on a TSA plate, incubated at 35 ± 2 °C, overnight. 5–10 well-isolated colonies of similar morphology were selected and re-streaked onto TSA plates and incubated at 35 ± 2 °C, overnight. 5 colonies from the overnight agar plate were inoculated into 30 mL of CAMHB in a sterile 125 mL flask and shaken at 35 ± 2 °C and 225 rpm to OD$_{600}$ ≈ 0.5 (i.e., mid-logarithmic phase). Cells were collected by centrifugation at $5000 \times g$ for 7 min, and washed once with assay buffer (5 mM HEPES pH 7.2, 20 mM glucose, 100 mM KCl). Cells were resuspended in the assay buffer to OD$_{600}$ ≈ 0.1. DiSC3(5) was added to the cells at final concentration of 100 nM, wrapped with aluminum foil, and incubated at 35 ± 2 °C, 220 rpm for 30 min until a stable reduction of fluorescence was observed. Two microliter of compound or DMSO were transferred into 96-well black walled polystyrene plates (Corning, Cat. No. 3603), to which was added 148 µL well$^{-1}$ of the dye-saturated bacterial cells to initiate the reaction. Plates were immediately read on an Envison at $\lambda_{ex}$ = 622 nm/ $\lambda_{em}$ = 670 nm and 35 °C with a kinetic program (reading every 10 s for 30 min)

until the maximal intensity was achieved and/or effect of positive control nisin was observed.

**Membrane permeability assay.** To evaluate the ability of the compounds to disrupt the cytoplasmic membrane integrity, the membrane impermeable fluorescent DNA intercalating dye Propidium iodide (PI) (Sigma Aldrich, Cat. No. P4864) was used. Early exponential phase cells were pelleted and washed twice in 10 mM HEPES buffer (pH 7.4) containing 50 µg mL$^{-1}$ of CaCl$_2$ and 5 mM glucose, and then re-suspended in the same buffer ($4 \times 10^8$ CFU mL$^{-1}$). Cell suspension was added to a 384-well black walled polystyrene plate (Corning, Cat. No. 3573) containing compounds, giving a final concentration of 16 µg mL$^{-1}$. After 1 h incubation at 37 °C, 5 µg mL$^{-1}$ of PI was added and fluorescence was monitored for 90 min using a Tecan Infinite® m1000 Pro Multi-mode reader (excitation/ emission 535/620 nm). Data were corrected by subtraction of fluorescence signal arising from untreated cells in the presence of PI. Each sample was tested in quadruplicate and independent assays were performed twice showing similar results. Data was analyzed and presented using Prism 8 software.

**Bacterial cytological profiling (Linnaeus Bioscience Inc.).** Samples of S. aureus ATCC 29213 or Bacillus subtilis PY79 were cultured in Luria-Bertani broth (LB) at 30 °C to an OD$_{600}$ ≈ 0.15. The cultures were then treated with 1×, 2×, or 5× MIC of CBD (5, 10, or 25 µg mL$^{-1}$). Samples (100 µL volume) were collected after 10, 30, and 120 min of CBD exposure and stained with 0.4 µM SYTOX™ Green (Invitrogen, Cat. No. S7020), 3 µg mL$^{-1}$ 4′,6-diamidino-2-phenylindole (DAPI; Roche Cat. No. 10236276001 purchased by Sigma Aldrich) and 1.5 µg mL$^{-1}$ of N-(3-triethy-lammoniumpropyl)—4-(6-(4-(diethylamino) phenyl) hexatrienyl) pyridinium dibromide (FM™ 4-64; Invitrogen, Cat. No. T3166). The samples were then centrifuged at ~10,000 rpm for 1 min, resuspended in ≈10 µL of the supernatant, and transferred to an agarose pad (20% LB, 1% agarose) for imaging.

**Macromolecular synthesis assay (HD Biosciences).** S. aureus RN4220 was cultured in CAMHB (BD, Cat. No. 212322) at 37 °C, diluted 1:100 in the same media and adjusted to OD$_{600}$ = ~0.05, then incubated at 37 °C to OD$_{600}$ = ~0.7. Then, 75 µL of CBD or reference compounds were serially diluted, with 5 µL ultimately transferred to corresponding columns of five 96-well tissue culture treated plates (BD Falcon, Cat. No. 353227). Radioisotope substrates (all from PerkinElmer) to detect DNA ([2-$^{14}$C]-Thymidine), RNA ([5,6-$^3$H]-Uracil), protein (L-[4,5-$^3$H]-Leucine), phospholipid ([2-$^3$H]-Glycerol), and peptidoglycan ([$^{14}$C(U)]-Glycine) were added to the appropriate wells, then 80 µL of the broth culture added into all except the blank wells. Plates were incubated at 35 °C for 25 min on a shaker plate (speed: 2–3). Then, 85 µL of the broth culture was added to the blank wells, and the bacteria precipitated by adding 100 µL ice cold TCA (12.5%), with plates set on ice for 60 min.

*Analysis of radioactivity.* UniFilter GF/B filter plates were pre-washed with 5% TCA in PerkinElmer Unifilter-96 Filtermate Cell Harvester. Bacterial samples of assay plates were filtered through filter plates. Filter plates were wash twice with 200 µL of ice-cold 5% TCA, once with 10% ethanol, then dried for at least 10 min. Plates were sealed, MicroScint-20 scintillation fluid (PerkinElmer) added, then counted on a TriLux MicroBeta or TopCount. Percent radiolabelling was calculated, curve fit performed, and half maximal effective concentrations calculated with GraphPad Prism 8.

**Cytotoxicity assay.** Cannabidiol and tamoxifen (Sigma Aldrich, Cat. No. T5648) in a 10 mg mL$^{-1}$ stock in 100% DMSO were prepared from powders then dried in a 96-well V bottom polypropylene (PP) plates (Greiner; M8185) using a Genevac solvent evaporator. Compounds were resuspended to 512 µg mL$^{-1}$ directly in cell culture medium then serially diluted two-fold for eight concentrations to give a final test concentration ranges of 256–2 µg mL$^{-1}$ and 64–0.5 µg mL$^{-1}$, respectively. HEK-293 cells (ATCC CRL-1573, human embryonic kidney epithelial cells) suspended in DMEM (Dulbecco's Modified Eagle's Medium; Gibco, Cat. No. 11330032) supplemented with 100 U mL$^{-1}$ Pen/Strep (Invitrogen, Cat. No. 15070063) and 10% fetal bovine serum (FBS, GE Healthcare, SH30084.03) were seeded into 384-well, black wall, clear bottom tissue culture (TC) plates (Corning, Cat. No. 3712) at 5000 cells per well in a volume of 20 µL. Manually, 20 µL of each compound dilution was plated in triplicate on the cells. The cell plates were incubated for 20 h at 37 °C, 5% CO$_2$.

Cytotoxicity (or cell viability) was measured by fluorescence, ex: 560/10 nm, em: 590/10 nm (F560/590), after addition of 5 µL of 100 µM resazurin (Sigma Aldrich, Cat. No. R7017) for a final concentration of approximately 11 µM, and after incubation for further 3 h at 37 °C in 5% CO$_2$. The fluorescence intensity (FI) was measured using a Tecan M1000 Pro monochromator plate reader, using automatic gain calculation.

The data was analyzed using Microsoft Excel and GraphPad Prism software. Cytotoxicity or cell viability were calculated using the following equation:

$$\text{Cell viability}(\%) = \left( FI_{sample} - FI_{negative} / FI_{untreated} - FI_{negative} \right) \times 100.$$

CC$_{50}$ (concentration at 50% cell viability) values were calculated using nonlinear regression analysis of log(concentration) vs. normalized cell viability, using variable fitting.

**Hemolysis assay**. Cannabidiol, along with hemolytic positive control melittin (Sigma Aldrich, Cat. No. M2272) were plated in U bottom, clear polypropylene 384-well plates (Corning, Cat. No. 3657) in a 2-fold, serial dilution with a final test concentration range of 256–2 µg mL$^{-1}$ and 32–0.25 µg mL$^{-1}$, respectively. Human whole blood (ARCBS; 00150) was washed three times in 0.9% NaCl (Baxter; AHF7124) and resuspended to a concentration of $5 \times 10^8$ cells mL$^{-1}$, as determined by manual cell count in a Neubauer haemocytometer. The washed cells were then added to the 384-well compound-containing plates for a final volume of 50 µL. After 10 min shaking on a plate shaker the plates were then incubated for 1 h at 37 °C in 5% CO$_2$. After incubation, the plates were centrifuged at $1000 \times g$ for 10 min to pellet cells and debris, 25 µL of the supernatant was then transferred to a flat bottom polystyrene 384-well assay plate (Corning, Cat. No. 3680).

Hemolysis was determined by measuring the supernatant absorbance at 405 nm (OD$_{405}$) using a Tecan M1000 Pro monochromator plate reader.

HC$_{10}$ and HC$_{50}$ (concentration at 10% and 50% hemolysis, respectively) were calculated by curve fitting the inhibition values vs. log(concentration) using a sigmoidal dose-response function with variable fitting values for top, bottom and slope. The maximal percentage of hemolysis is reported as DMax. The curve fitting was implemented using Pipeline Pilot's dose-response component, resulting in similar values to curve fitting tools such as GraphPad's Prism v8 and IDBS's XlFit.

The use of human blood (sourced from the Australian Red Cross Blood Service) for hemolysis assays was approved by the University of Queensland Institutional Human Research Ethics Committee, Approval Number 2014000031.

**Ex-vivo pig skin assay (Perfectus Biomed, formerly Extherid Biosciences)**[39]
*Explant preparation*. Porcine tissue (from *Sus scrofa domesticus*), transported on ice, were received 2–5 h after slaughter. In Rosewell Park Memorial Institute (RPMI) 1640 medium (Gibco™, Cat. No. 11875085) containing 2% (v/v) penicillin/streptomycin, a 5 mm biopsy punch was used to cut tissue explants and remaining muscle tissue removed with a sterile scalpel blade. Tissue was antibiotic treated (for decontamination of flora) for $0.5 \pm 0.25$ h. Explants were rinsed three-times with $10 \pm 0.5$ mL RPMI (no antibiotic, no FBS). Explants were then covered with fresh RPMI (no antibiotic, no FBS) and placed at $4 \pm 2$ °C for $12 \pm 4$ h (antibiotic washout). Overnight RPMI was then removed and replaced with $10 \pm 0.5$ mL fresh RPMI $15 \pm 5$ min prior to infection.

*Bacterial inoculation*. Fresh plates were streaked directly from frozen stock within 3 weeks of the experiment. Culture tubes containing Todd Hewitt broth (THB) were inoculated with a single colony and placed in a shaking incubator (at $37 \pm 2$ °C, $150 \pm 10$ rpm) late afternoon the day before the experiment. On the morning of the experiment, $200 \pm 50$ µL of overnight culture was transferred into $2 \pm 0.5$ mL fresh THB and shaken for $3 \pm 1$ h at 37 °C. Inoculum was then washed to a final concentration of $5 \times 10^8$ CFU mL$^{-1}$.

*Model set up*. 6-well plates were set up with $2 \pm 0.2$ mL RPMI (no antibiotic, no FBS) in each well and a 0.4 µm trans-well insert. Tissue explants were transferred into wells mucosal side up to the insert.

*Infection and treatment*. Explants were infected with $2 \pm 0.5$ µL of prepared inoculum (approximately $1 \times 10^6$ CFU/explant or $5 \times 10^8$ CFU mL$^{-1}$). Explants were incubated at $37 \pm 2$ °C for $2 \pm 0.5$ h, then treatments (12 CBD-containing formulations and associated vehicles) administered in triplicate and incubated for at $37 \pm 2$ °C for $1 \pm 0.25$ h.

*Wash*. Post-treatment, $1.0 \pm 0.05$ mL sterile phosphate-buffered saline (PBS) + 2% (w/v) mucin was added to each insert for the appropriate tissue and swirled gently for 5 sec. The liquid suspension was then aspirated, and wells replenished with RPMI ($2 \pm 0.2$ mL RPMI [no antibiotic, no FBS]). Explants were returned to the 37 °C incubator for the indicated post-treatment timepoints: $1.0 \pm 0.25$ h, $24 \pm 4$ h.

*Sample collection*. Post-wash ($1.0 \pm 0.25$ h, $24 \pm 4$ h), tissue was removed from trans-wells and placed in $500 \pm 0.03$ µL of neutralizer (30 mg mL$^{-1}$ bovine serum albumin). Samples were sonicated and vortexed ($30 \pm 5$ s vortex, $120 \pm 6$ s sonicate, $30 \pm 5$ s vortex). Samples were then plated neat or diluted in sterile PBS. $50 \pm 2$ µL of sample was plated with a spiral plater on mannitol salt agar, and plates incubated for 24–48 h at $37 \pm 2$ °C. The following day, colonies were counted with an automated plate counter and CFU counts transformed to Log$_{10}$(CFU/explant).

**Bioluminescent in vivo mouse skin infection model (Charles River Laboratories)**. Mice ($n = 6$ per group, adult female *CD1* mice) were allocated to receive vehicle, Bactroban (2% mupirocin, 50 µL), or one of 3 formulations of CBD (BTX 1503, 1503 gel and BTX 1204 gel, 50 µL) topically 0, 12, 24, and 32 h post-infection.

*S. aureus* Xen-29 bioluminescent bacteria was cultured under standard conditions. On Days −4 and −1, mice were administered with 150 then 100 mg kg$^{-1}$ cyclophosphamide, respectively. On Day 0, an area of the back was shaved, and the skin surface disrupted to facilitate bacterial colonization. The area was then inoculated with $5 \times 10^7$ CFU of bacteria in a 10 µL droplet and the droplet spread around the area. On Day 0 animals received their first treatment at 0 h post-infection and subsequent treatments at 12, 24, and 32 h after infection. Bioluminescent in vivo imaging was performed pre-infection then at 4, 24, 36, and 48 h post-infection. On Day 2, 48 h after the first treatment and infection, animals were sacrificed, and the skin assessed for signs of gross pathology. Samples of skin from each animal were homogenized and plated on nutrient agar in order to determine the CFU's per animal. Animals were imaged using a Lumina II system (Perkin-Elmer), and bioluminescence (photons per second) determined.

**MRSA thigh infection in vivo efficacy model**. This assay was performed at the University of Queensland, following similar previous literature procedures[67].

*Compound preparation*. Cyclophosphamide monohydrate (Sigma Aldrich, Cat No. C0768) was dissolved in sterile saline to a concentration of 30 mg mL$^{-1}$. Likewise, vancomycin was also dissolved in sterile saline to a final concentration of 60 mg mL$^{-1}$.

*Preparation of injectable MRSA solution*. MRSA bacterial isolate was taken from the storage at −80 °C and freshly seeded on agar plates for overnight growth. From the overnight culture preparation, a single colony was diluted into 5 mL of MHB and incubated overnight at 37 °C. A log-phase subculture was obtained by adding 150 µL of overnight subculture in 10 mL MHB and incubated for a further 2–3 h. Finally, the OD$_{600}$ value of the bacterial suspension was determined and the colony forming units per milliliter (CFU mL$^{-1}$) extrapolated. A full dilution of the bacterial cell suspension in saline was achieved by washing ($3220 \times g$ for 10 min) and the OD$_{600}$ in saline determined. The suspension was then diluted out accordingly in order to achieve a $2 \times 10^6$ CFU mL$^{-1}$ solution ($10^5$ CFU in 50 µL thigh$^{-1}$).

*Quantification of injected MRSA solution*. In order to be able to correlate the actual CFU mL$^{-1}$ present in the MSRA injection solution with the estimated CFU mL$^{-1}$ based on the OD$_{600}$ readings, a standard plate count from the MSRA injection solution was performed. Thus, 10 µL of the injectable MRSA suspension was diluted down to 10-fold to 1000-fold, and each dilution plated out onto agar and incubated at 37 °C for 24 h. From the estimated $2 \times 10^6$ CFU mL$^{-1}$ solution, 18 CFU per 10 µL were found in the 1:1000 dilution, giving a final concentration of $1.8 \times 10^6$ CFU mL$^{-1}$ for the actual injectable MRSA solution.

*In vivo experimental assay*. Seven-week-old female outbred *CD1* mice (UQBR-AIBN) were rendered neutropenic by injecting two doses of cyclophosphamide intraperitoneally 4 days (150 mg kg$^{-1}$) and 1 day (100 mg kg$^{-1}$) prior to experimental infection. The infection model using MRSA was established by intramuscular injection of 50 µL of early-log-phase bacterial MRSA suspension (around $2 \times 10^6$ CFU mL$^{-1}$) in saline into both thigh muscles. Two hours later, a single dose of vancomycin (200 mg kg$^{-1}$) was administered by a subcutaneous injection over the interscapular (area at back of the neck), and CBD was administered by subcutaneous, intraperitoneal or oral dosing. Untreated animals received equivalent volume of saline (CES). The mice were monitored for signs of normal behavior (i.e., grooming, eating, drinking, sleeping, and alertness, movement, etc.) during and following dosing. Twenty-six hours after MRSA infection, mice were euthanised. For each mouse, both thighs were collected aseptically by cutting the leg at the hip and knee, placed in 10 mL of cold sterile saline and the individual weight of each thigh recorded.

*Thigh homogenates and CFU determination*. Thighs were homogenized at 20,000 rpm for 15 s using a Polytron MR2500E using a 200 mm probe (Kinematica). Homogenate solutions were filtered using a 100 µm pore size filter (BD) and 1 mL of filtrate solution placed on ice and serial dilutions promptly done and seeded onto appropriate nutrient agar plates (Bactolaboratories) and incubated at 37 °C overnight. Colonies were counted the next day and CFU thigh$^{-1}$ and the CFU g$^{-1}$ of thigh calculated based on the plate count and dilution factor.

**Statistics and reproducibility**. The statistical tests and number of biological replicates and/or experiments are stated in the figure subtexts, with significance set at $p < 0.05$. Statistical analysis was done using Graphpad Prism 8.

**Reporting summary**. Further information on research design is available in the Nature Research Reporting Summary linked to this article.

## Data availability
All relevant data are available in this article and its Supplementary Information files, except for original image files, which are available from the corresponding author upon reasonable request. All data underlying the graphs is available as Supplementary Data.

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

## Acknowledgements
This work was supported by the Australian Department of Industry, Innovation and Science AusIndustry Innovation Connections Grants ICG000601 and ICG001154, with funding from Botanix Inc. M.A.B, A.K., S.R., G.L., A.G.E., M.A., A.O.H. and J.Z. were supported in part by Wellcome Trust Strategic Grant 104797/Z/14/Z and B.Z. by a CSC Scholarship. The following strains were provided by the Network on Antimicrobial Resistance in *Staphylococcus aureus* (NARSA) for distribution by BEI Resources, NIAID, NIH: *S. aureus* NRS-1, and VRS-4. ATCC-labeled strains and human cell lines HepG2 (ATCC HB-8065) and HEK-293 (ATCC CRL-1573), were acquired from American Type Culture Collection (ATCC). We thank Prof Graeme Nimmo, Queensland Health Central Pathology, for clinical isolates of *S. aureus*. The *P. aeruginosa* wild type strain PAO1, and the isogenic multiple Mex pump mutant, PAO750 (Δ(mexAB-oprM), Δ(mexCD-oprJ), Δ(mexEF-oprN), Δ(mexJK), Δ(mexXY), ΔopmH, ΔpscC), were a gift from Herbert Schweizer at Colorado State University[68]. Microscopy was performed at the Australian Cancer Research Foundation (ACRF)/Institute for Molecular Bioscience Cancer Biology Imaging Facility, which was established with the support of the ACRF. We thank the Australian Red Cross Blood Service for the supply of blood for hemolysis assays.

## Author contributions
M.A.T.B. designed the project with input from M.T. and M.C. M.A.T.B. wrote the manuscript with substantial input from A.M.K. and A.G.E. and review by all authors. M.A.T.B., M.T., and A.M.K. designed and coordinated experiments and analyzed results. M.A.T.B designed and N.B. synthesized, purified and analyzed compounds. A.M.K. conducted most UQ-based microbiological experiments, with assistance from A.G.E., S.R., M.A., G.J.L., A.O.H., D.M.T.P., and J.Z. B.Z. conducted biofilm microscopy experiments, D.Q., M.D.S., and J.P. conducted cytological profiling experiments, L.T. and N.W. conducted *Mtb* testing, A.P.R. and D.L. conducted *C. difficile* testing, and D.W.C, and M.L.P. carried out ex vivo pig skin colonization assays and analysis.

## Competing interests
The authors declare the following competing interests: M.C. and M.T. are employees of Botanix Inc, who co-funded most of this research. M.A.T.B. consults for Botanix Inc., and is an inventor on several antibiotic patents (unrelated to this work) which are undergoing commercialization. D.Q. and M.D.S. are employees and J.P. founded Linneas (conducted the bacterial cytological profiling). J.P. has an equity interest in Linnaeus Bioscience Incorporated and receives consulting income from the company. The terms of this arrangement have been reviewed and approved by the University of California, San Diego in accordance with its conflict of interest policies. N.B. is founder of BDG Synthesis (conducted the analog synthesis), and D.W.C. is an employee and M.L.P. founded Extherid Biosciences (conducted the ex-vivo pig skin studies), which is now part of Perfectus Biomed Group. The remaining authors declare no competing interests.
