## [Peer Review File · Communications Biology]

Editorial Note: This manuscript has been previously reviewed at another journal that is not operating a transparent peer review scheme. This document only contains reviewer comments and rebuttal letters for versions considered at Communications Biology.

Reviewers' comments:

Reviewer #1 (Remarks to the Author):

I was one of the previous Reviewers. I am satisfied with the revised paper and am hopeful that the authors can further pursue the work to develop a novel antimicrobial. My only concern is that the anti-gonococcal MIC testing using both agar and microbroth dilution while giving similar results represent an N=1 for both assays. At the very least the authors should select a few strains and perform replicate experiments.

Reviewer #2 (Remarks to the Author):

The manuscript by Blaskovick and co-workers describes an extensive evaluation of CBD both in vitro and in vivo, as well as preliminary mechanistic studies and preliminary analog studies. The main findings are that:

1. CBD is active against Gram-positive organisms
2. CBD has limited activity against Gram-negative, but notably has activity against Neisseria
3. CBD is able to inhibit biofilm formation at its MIC concentration
4. Mechanistic studies, including macromolecular synthesis assays and membrane permeabilization assays indicate that membrane disruption is driving activity, although no target is identified
5. The authors show that 2 of the 12 formulations they employ show high efficacy in an ex vivo porcine model in comparison to vehicle formulation
5. CBD is active in vivo against a mouse topical infection model, although not as effective as the positive control mupirocin
6. CBD was not active in a thigh infection model when CBD was delivered systemically
7. Preliminary analog synthesis, to ultimately reduce protein binding, generated new analogs with a modest improvement of activity and a preliminary dissection of what functional groups drive activity against MRSA as opposed to Neisseria.

Overall, I found the information in the manuscript to be highly impactful and well presented. I only have a couple of minor comments to the authors, both on page 8:

1. The authors describe a synergy experiment with CBD and polymyxin B and colistin to show that disruption of the outer membrane potentiates CBD activity. I would suggest two experiments to remove the ambiguity of the antibiotic activity driven by either polymyxin. First, repeat the experiment with Polymyxin B nanapeptide (PMBN). This will permeabilize Gram negative bacteria but not act as an antimicrobial agent. If the authors hypothesis is correct (i.e. CBD lacks penetration to most gram negative pathogens), then PMBN should lower the CBD MIC to levels around the values they observe for gram positive strains. I would follow this up by generating the LPS deficient mutant of *A. baumannii* and test CBD activity. LPS deficient *A. baumannii* is easily isolated by plating a culture on plates containing high levels of colistin and using the resulting colonies.

2. The MBEC assay the authors describe is not an MBEC assay. An MBEC assay traditionally uses the Calgary device where you first generate biofilms on the pegs for a set time (usually 24-48 hours), remove the pegs from the bacterial inoculum, wash off loosely attached bacteria, and then add them to a new plate containing compound in solution. After 24 hours, the pegs are removed, washed, and then placed in sterile media. After incubation for 16-24 hours, the MBEC is recorded as the lowest concentration of test article that stops outgrowth from the PEGS. What the

authors describe is more of a biofilm inhibition study. The data they present is fine, its just not a typical MBEC assay and shouldn't be referred to as such.

Manuscript COMMSBIO-20-1951A “The antimicrobial potential of cannabidiol”.

Response to Reviewers:

Reviewer #1:

I was one of the previous Reviewers. I am satisfied with the revised paper and am hopeful that the authors can further pursue the work to develop a novel antimicrobial. My only concern is that the anti-gonococcal MIC testing using both agar and microbroth dilution while giving similar results represent an N=1 for both assays. At the very least the authors should select a few strains and perform replicate experiments.

We conducted additional replicate testing for *entire* panel using the agar dilution gonococcal MIC procedure (previous reviewers had recommended this assay method was preferred over the broth microdilution assay as it is the CLSI recommended assay for *N. gonorrhoeae*). We now have three independent replicates against the entire panel. Values were generally consistent for CBD (MIC_{50/90} slightly elevated from 1/2 to 2/ 4 µg/mL) and other antibiotics. The discussion has been slightly amended to reflect these results.

In addition, for the broad panel testing conducted at Micromyx (Supplementary table 2, superscripted ‘2’ results), we also conducted 2 additional replicates to give n =3, even though this was not requested.

Reviewer #2:

Overall, I found the information in the manuscript to be highly impactful and well presented. I only have a couple of minor comments to the authors, both on page 8:”

*1. The authors describe a synergy experiment with CBD and polymyxin B and colistin to show that disruption of the outer member potentiates CBD activity. I would suggest two experiments to remove the ambiguity of the antibiotic activity driven by either polymyxin. First, repeat the experiment with Polymyxin B nonapeptide (PMBN). This will permeabilize Gram negative bacteria but not act as an antimicrobial agent. If the authors hypothesis is correct (i.e. CBD lacks penetration to most gram negative pathogens), then PMBN should lower the CBD MIC to levels around the values they observe for gram positive strains. I would follow this up by generating the LPS deficient mutant of *A. baumannii* and test CBD activity. LPS deficient *A. baumannii* is easily isolated by plating a culture on plates containing high levels of colistin and using the resulting colonies.*

These were excellent suggestions, and we conducted both experiments. Polymyxin B nonapeptide and CBD alone were inactive against *E. coli*, *K. pneumoniae*, *A. baumannii* and *P. aeruginosa* (MIC > 32 µg/mL and >256 µg/mL, respectively). When used in combination, they showed synergy against 3 strains (n=4), though not against *K. pneumoniae*. The CBD MIC was reduced to 32-64 µg/mL with as little as 2-4 µg/mL of PMBN.

Rather than generating a LPS-deficient µg/mL *A. baumannii* strain, we used the colistin resistant strain *A. baumannii* AL1851, an lpxA mutant that is Lipid A deficient, as described by Moffat et al. “Colistin resistance in *Acinetobacter baumannii* is mediated by complete loss of lipopolysaccharide production.” *Antimicrob Agents Chemother*, 2010 54, 4971-4977. doi:10.1128/AAC.00834-1. We also tested the parent strain from which it is derived, ATCC 19606. In agreement with Moffat, in our assay the polymyxin-resistant LPS-deficient strain gained susceptibility to teicoplanin and gentamicin, providing indirect confirmation that the strain tested is indeed LPS-deficient. The CBD MIC decreased from >128 µg/mL in ATCC19606 to 0.25 µg/mL in AL1851, confirming the role of LPS in preventing CBD Gram-negative activity.

These experiment are now discussed in the results section and conclusion, and experimental detail added to the methods section.

Additional data:

We have also added in some additional MIC testing results:

- 1) activity vs VRE (Table 1 and Supplementary Table 2)
- 2) MIC₉₀ vs an additional 100 clinical isolates of *Staphylococcus aureus* from a different geographical location and 60 beta-hemolytic streptococci (new Supplementary Tables 3-5). Please note that while these new results are only n=1 for each strain, this is industry standard for MIC₉₀ testing.

These experiment are now discussed in the results section, and experimental detail added to the methods section. Figure 1 has been modified to include the additional MIC₉₀ data.

REVIEWERS' COMMENTS:

Reviewer #2 (Remarks to the Author):

The authors have addressed the concerns from the previous round of reviews. I am happy to recommend publication.